# Actin networks modulate heterogeneous NF-κB dynamics in response to TNFα

Francesca Butera[1]*[†], Julia E Sero[2], Lucas G Dent[1], Chris Bakal[1]*

[1]Chester Beatty Laboratories, Division of Cancer Biology, Institute of Cancer Research, London, United Kingdom; [2]Department of Life Sciences, University of Bath, Bath, United Kingdom

**Abstract** The canonical NF-κB transcription factor RELA is a master regulator of immune and stress responses and is upregulated in pancreatic ductal adenocardinoma (PDAC) tumours. In this study, we characterised previously unexplored endogenous RELA-GFP dynamics in PDAC cell lines through live single-cell imaging. Our observations revealed that TNFα stimulation induces rapid, sustained, and non-oscillatory nuclear translocation of RELA. Through Bayesian analysis of single-cell datasets with variation in nuclear RELA, we predicted that RELA heterogeneity in PDAC cell lines is dependent on F-actin dynamics. RNA-seq analysis identified distinct clusters of RELA-regulated gene expression in PDAC cells, including TNFα-induced RELA upregulation of the actin regulators NUAK2 and ARHGAP31. Further, siRNA-mediated depletion of ARHGAP31 and NUAK2 altered TNFα-stimulated nuclear RELA dynamics in PDAC cells, establishing a novel negative feedback loop that regulates RELA activation by TNFα. Additionally, we characterised the NF-κB pathway in PDAC cells, identifying how NF-κB/IκB proteins genetically and physically interact with RELA in the absence or presence of TNFα. Taken together, we provide computational and experimental support for interdependence between the F-actin network and the NF-κB pathway with RELA translocation dynamics in PDAC.

*For correspondence:
frankie.butera@mcri.edu.au (FB);
cbakal@icr.ac.uk (CB)

Present address: [†]Murdoch Children's Research Institute, The Royal Children's Hospital, Melbourne, Australia

Competing interest: The authors declare that no competing interests exist.

## Editor's evaluation

This paper presents an important investigation of the relationship between cell morphology, actin cytoskeletal features, and NF-kappaB/RELA signaling dynamics. Solid evidence is provided using quantitative live-cell imaging of pancreatic cancer cell lines. These analyses better establish the connection of cell shape to the NF-kappaB signaling pathway, highlighting the importance of the actin network and several specific regulators in a feedback loop controlling NF-kappaB activity. Because NF-κB controls inflammation and cell survival, this study will be of interest in the fields of cancer and immune signaling.

## Introduction

The NF-κB transcription factor RELA is an essential mediator of the inflammatory and immune responses in all mammals (**Hayden et al., 2006**) and is central to the canonical NF-κB signalling pathway (**Ghosh et al., 1998**). As a transcription factor, RELA activation is controlled in large part through its localisation. Inactive RELA is sequestered in the cytoplasm by IκB proteins and IκB degradation by upstream cues, such as the potent inflammatory cytokine tumour necrosis factor α (TNFα), enables RELA translocation to the nucleus where RELA regulates gene expression (**DiDonato et al., 1997**; **Zandi et al., 1997**).

Live imaging experiments of fluorescently labelled RELA and electrophoretic mobility shift assays have shown that RELA oscillates between the nucleus and cytoplasm in response to TNFα (**Hoffmann**

**eLife digest** The prognosis for the most common type of pancreatic cancer, pancreatic ductal adenocarcinoma, also known as PDAC, remains poor. Only around 4% of PDAC patients are likely to live 5 years after being diagnosed. The immune system plays a part in the progression of PDAC, as increased inflammation contributes to the growth of the tumour and its ability to resist treatment.

The NF-κB proteins of the immune system are transcription factors that control when and how much certain other proteins are produced. The protein RELA is an important member of the NF-κB family. It is known to be overactivated in PDAC tumours and may be responsible for the high levels of inflammation found in this type of pancreatic cancer. A better understanding of how RELA is activated in PDAC cells will help develop medicines targeting this process.

Butera et al. combined experimental and computational approaches to model a network of interactions between key molecules in lines of human PDAC cells grown in the laboratory to investigate how RELA is controlled. Usually, when the RELA protein is activated, it is located in the cell's nucleus, this means that tracking the location of RELA within the cell can reveal its activated form. To follow RELA, it was labelled with a fluorescent marker and visualised using live, high-throughput fluorescence microscopy.

When RELA was activated with a pro-inflammatory molecule TNFα, it changed how it moved in and out of the cell's nucleus. Using computational modelling approaches, Butera et al. could build a statistical model that revealed that the location and activity of RELA is affected by actin, a key component of the cell's molecular scaffolding called the cytoskeleton.

When actin was disrupted, RELA's activity was altered. Moreover, profiling targets of RELA using RNA sequencing revealed two genes that encode proteins related to the regulation of actin. This indicates that RELA activity – which is itself affected by actin organisation – can feed back to regulate actin.

Butera et al. have identified genes regulated by RELA in PDAC cells that could be used as targets for anti-cancer drug development. Further research into the feedback between RELA and actin will help untangle the complex network that causes inflammation in pancreatic cancer.

*et al., 2002*; *Nelson et al., 2004*; *Sung et al., 2009*; *Tay et al., 2010*; *Sero et al., 2015*; *Zambrano et al., 2016*). Oscillations are driven by a negative feedback loop between RELA and particular IκB isoforms since the genes encoding IκB proteins are RELA transcriptional targets (*Brown et al., 1993*; *Scott et al., 1993*; *Sun et al., 1993*; *Hoffmann et al., 2002*). The pattern of RELA translocation has been shown to dictate the specificity and timing of RELA target gene expression, including the genes encoding IκBα, IκBε, and the chemokine RANTES (*Ashall et al., 2009*; *Zambrano et al., 2016*; *Lane et al., 2017*).

Previously, we showed that cell shape is a regulator of RELA dynamics in breast cancer cell lines and that breast cancer cells with mesenchymal cell shape (protrusive with low cell–cell contacts) have higher RELA nuclear translocation (*Sero et al., 2015*). We also identified that RELA activity, coupled to cell shape, is predictive of breast cancer progression (*Sailem and Bakal, 2017*). Although the mechanistic basis for how cell shape regulates RELA remains poorly understood, studies have shown that chemically inhibiting actin or tubulin dynamics can increase RELA binding to DNA and RELA-dependent gene expression (*Rosette and Karin, 1995*; *Bourgarel-Rey et al., 2001*; *Németh et al., 2004*).

Despite frequent upregulation of both TNFα and RELA in PDAC tumours (*Weichert et al., 2007*; *Zhao et al., 2016*), the dynamics and regulation of single-cell RELA translocation, as well as RELA transcriptional output, are poorly understood for PDAC cells. Here, we used CRISPR-CAS9 to tag RELA with GFP in the human PDAC cell lines MIA PaCa2 and PANC1 and identified rapid, non-oscillatory, and heterogeneous RELA dynamics by live imaging. To explore potential cytoskeletal or cell shape regulation of RELA in PDAC, we constructed Bayesian models using single-cell datasets from TNFα-stimulated PDAC cells with variation in RELA, actin, and tubulin measurements for hypothesis generation: one dataset with five PDAC cell lines and another with diverse small molecules targeting cytoskeletal components. Notably, nuclear RELA was statistically predicted as dependent on actin features across Bayesian models. Finally, we used RNA-seq to identify genes with distinct

patterns of expression based on dependence on TNFα dose and RELA activation. Using live imaging with knockdown of selected targets, our results uncover novel mechanisms regulating inflammation-associated RELA dynamics in PDAC, including negative feedback loops between RELA and the actin modulators NUAK2 and ARHGAP31, and between RELA and the non-canonical NF-κB protein RELB.

## Results

### Single-cell endogenous RELA responses to TNFα are non-oscillatory and sustained in PDAC cells

To study PDAC biology, we used the frequently studied human PDAC cell lines MIA PaCa2 and PANC1, which harbour key genomic alterations in PDAC, including KRAS and p53 mutations and homozygous deletions in CDKN2A/p16 (*Deer et al., 2010*). In addition, MIA PaCa2 and PANC1 cells are epithelial in origin but have distinct cell morphologies and are therefore useful for analysis of cell shape interaction with RELA. To study dynamic RELA localisation changes, we used CRISPR-CAS9 gene editing to fluorescently tag endogenous RELA at the C-terminus with eGFP (abbreviated as RELA-GFP) (*Figure 1A and B*). The C-terminus was selected to avoid interference with the N-terminal Rel homology domain, which contains the transactivation, nuclear import and DNA-binding domains (*Kieran et al., 1990*; *Nolan et al., 1991*). GFP+ clones were selected using FACS, single-cell sorted and expanded into monoclonal cell lines. We also introduced mScarlet-I to the C-terminus of PCNA (proliferating cell nuclear antigen; abbreviated as PCNA-Scarlet) – a processivity factor for DNA polymerase δ that functions during replication – which served as a nuclear marker for segmentation and as a cell cycle marker (*Kurki et al., 1986*; *Barr et al., 2017*; *Figure 1A and B*).

To observe live RELA translocation dynamics in response to inflammatory stimuli, we used timelapse confocal microscopy with automated image analysis to track changes in RELA-GFP localisation on a single-cell level in response to TNFα (0.01, 0.1, and 10 ng/ml), from –120 min to +600 min relative to TNFα addition. 0.01 ng/ml TNFα is a physiological dose relevant to healthy and malignant tissue, while 0.1 ng/ml TNFα is detected in highly inflammatory PDAC microenvironments (*Zhao et al., 2016*). 10 ng/ml TNFα was used in several studies assaying RELA translocation (*Hoffmann et al., 2002*; *Tay et al., 2010*; *Sero et al., 2015*) and is included for comparison, but is substantially above physiological levels (*Zhao et al., 2016*).

Broadly, we observed a lack of oscillatory nuclear RELA dynamics in PDAC cells, in addition to general maintenance of nuclear RELA across the 10 hr imaging period. MIA PaCa2 cells responded with higher nuclear RELA compared to PANC1 cells at each TNFα dose (*Figure 1C*). Peak nuclear RELA, quantified as the first timepoint at which the slope of the single-cell RELA track $\leq 0$, was statistically significantly different between 0.01 and 0.1 ng/ml for MIA PaCa2 but not for PANC1 cells, suggesting that MIA PaCa2 cells are more sensitive to TNFα dose in terms of RELA activation (*Figure 1D*). 10 ng/ml TNFα elicited a significantly higher nuclear RELA localisation amplitude in both cell lines compared to lower TNFα doses (0.01 and 0.1 ng/ml). In contrast, TNFα dose did not affect the time to peak RELA in MIA PaCa2 cells, while PANC1 cells displayed a slower time to peak at 0.01 ng/ml TNFα compared to higher doses (*Figure 1E*).

We also compared RELA measurements within each single-cell track to assess the stability of nuclear RELA levels (*Figure 1F*). In MIA PaCa2 cells, there is a high correlation between nuclear RELA at later timepoints (180, 300, or 600 min) compared to an early timepoint (60 min) following TNFα addition. PANC1 cells have high correlation between nuclear RELA measurements at 60 min RELA and 180 min, while there is weak correlation between RELA levels at 60 and 600 min. Thus, in MIA PaCa2, but not PANC1, RELA translocation at 60 min is largely predictive of the long-term response to TNFα and MIA PaCa2 cells maintain TNFα-induced nuclear RELA levels for an extended duration (10 hr) following exposure.

Overall, our data show that RELA nuclear translocation occurs rapidly, and with non-oscillatory dynamics in both MIA PaCa2 and PANC1 cells in response to TNFα. Nuclear RELA translocation is largely sustained for hours in MIA PaCa2 cells, while damped in PANC1 cells.

### RELA translocation responses to TNFα are cell cycle independent

To identify whether cell cycle progression contributes to heterogeneity in RELA responses to TNFα in PDAC cells, we categorised each tracked cell by cell cycle stage at the time of TNFα addition, using

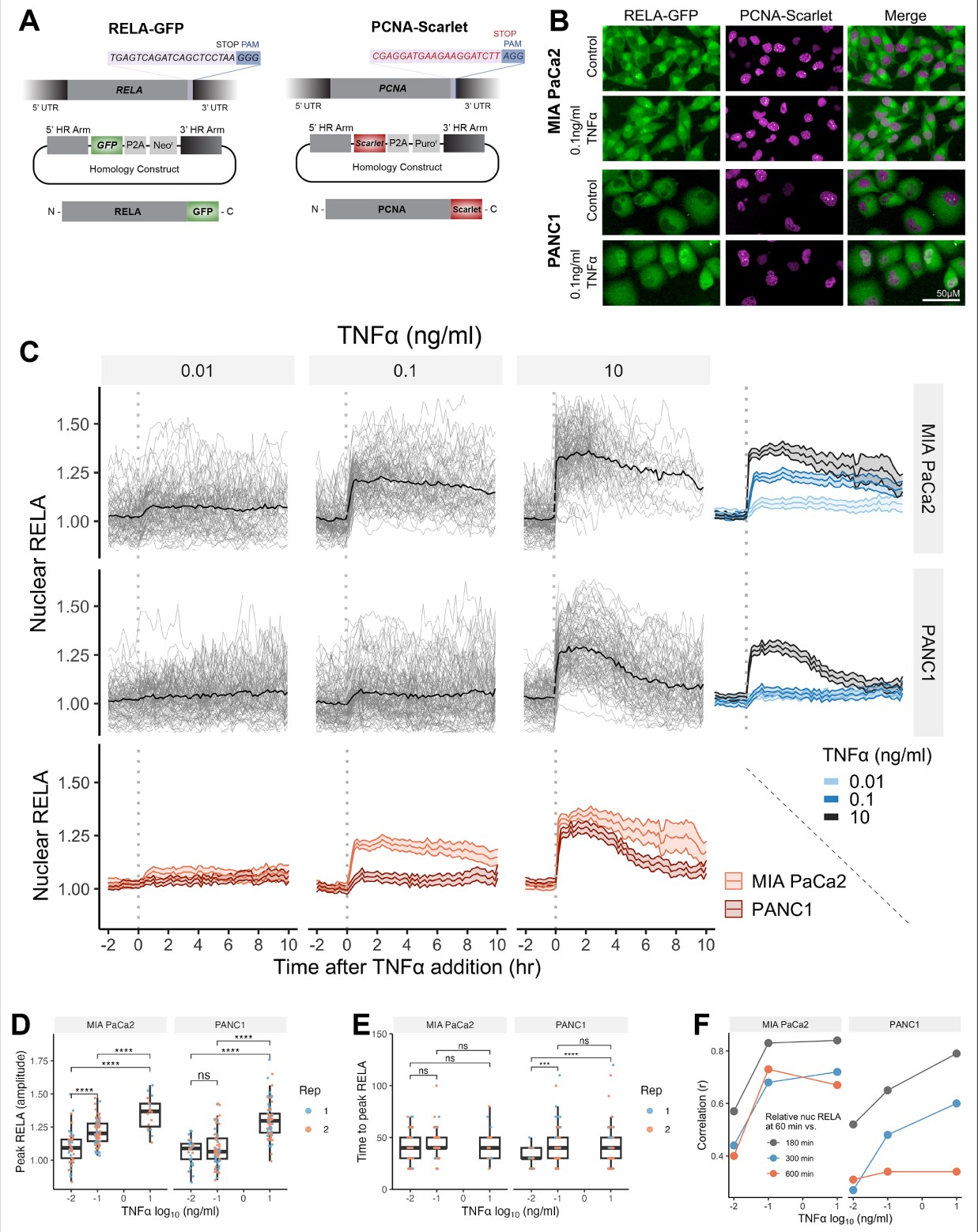

**Figure 1.** Single-cell analysis of live endogenous RELA-GFP dynamics with TNFα in human pancreatic cell lines. (**A**) Schematic of CRISPR gRNA and homology constructs, with neomycin (neo) resistance for tagging of the RELA C-terminus with GFP and puromycin (puro) resistance for tagging of the PCNA C-terminus with Scarlet. (**B**) Confocal microscopy images of MIA PaCa2 and PANC1 monoclonal cell lines expressing endogenously tagged RELA-GFP and PCNA-Scarlet and treated with 1 hr solvent control or 0.1 ng/ml TNFα. (**C**) Tracks of single-cell nuclear RELA-GFP intensity measurements in MIA PaCa2 and PANC1 cells from –120 min to +600 min relative to TNFα addition (0.01, 0.1, and 10 ng/ml). n = 50–60 tracked cells per TNFα dose

*Figure 1 continued on next page*

*Figure 1 continued*

for each of the two experimental repeats. (**D**) Amplitude and (**E**) time of first nuclear RELA-GFP intensity peak in MIA PaCa2 cand PANC1 single cells. Boxplots show median and interquartile range. M = median per cluster. σ = standard deviation. Statistical significance shown for *t*-tests with Benjamini–Hochberg correction. ns (non-significant) = p>0.05, *p<0.05, **p<0.01, ***p<0.001, ****p<0.0001. (**F**) Pearson's correlation coefficient (r) between nuclear RELA-GFP intensity at 60 min versus 180, 300, or 600 min within single-cell RELA-GFP tracks. Data are normalised within each track to nuclear RELA-GFP intensity at 0 min.

The online version of this article includes the following source data and figure supplement(s) for figure 1:

**Source data 1.** Excel file with raw data and data replication information associated with *Figure 1* and *Figure 1—figure supplement 1*.

**Figure supplement 1.** Cell cycle independence of TNFα-mediated RELA dynamics in PDAC cells.

changes in the appearance and intensity of endogenous PCNA-Scarlet to mark cell cycle transitions (*Figure 1—figure supplement 1A and B*). For each cell, we calculated the amplitude and timing of peak nuclear RELA, in addition to the mean nuclear RELA intensity across all timepoints. Largely, there were minimal differences between cells by cell cycle stage in terms of peak RELA measurements (amplitude or timing) in MIA PaCa2 and PANC1 cells (*Figure 1—figure supplement 1C and D*), with the exception of statistically significant lower peak RELA amplitude detected in G2 cells compared to S phase cells only, and exclusively for the MIA PaCa2 cell line at 10 ng/ml TNFα. Conversely, we identified statistically significant higher (whole track) mean nuclear RELA in G2 cells in the PANC1 cell line alone in response to 10 ng/ml TNFα (*Figure 1—figure supplement 1E*). Overall, nuclear RELA translocation in response to TNFα in PDAC cells appears to be largely cell cycle independent.

## TNFα-mediated RELA heterogeneity in PDAC cells is predicted to be dependent on actin dynamics

Because the same TNFα concentration can lead to variable responses, we proposed that there are cell-intrinsic mechanisms that dictate the extent of RELA translocation in PDAC cells. Having previously identified relationships between cell shape and RELA localisation in breast cells (*Sero et al., 2015*), we hypothesised that differences in actin and tubulin organisation, which regulate cell shape (*Machesky and Hall, 1997*, *Desai and Mitchison, 1997*), may explain differences in RELA dynamics. To test this, we expanded our dataset to include immunofluorescence images of the human immortalised PDAC cell lines MIA PaCa2, PANC1, Capan1, SW1990, and PANC05.04. We treated cells with TNFα (1 hr), or with solvent control, and stained for DNA, RELA, F-actin, and α-tubulin (*Figure 2—figure supplement 1A*). We used automated image analysis to segment cell regions and measured 35 geometric, cyto-skeletal, and Hoechst features, as well as nuclear RELA, in approximately 130,000 cells. The 35 cell features were then reduced by hierarchical clustering to a subset of 10 features (*Figure 2—figure supplement 1B*). Selected features include classical measurements of cell shape ('cell area', 'cell roundness', and 'nucleus roundness'), mean measurements and texture analysis of actin and tubulin intensity in the cytoplasm, and the ratio of 'actin filament area' to 'cell area' which assays actin stress fibre abundance (*Figure 2—figure supplement 1C*).

We used principal component analysis (PCA) to assess the morphological diversity of the five PDAC cell lines using normalised data for cells under control conditions (*Figure 2—figure supplement 2*). PCA largely clustered data by cell line, indicating distinct cell morphology and cytoskeletal organ-isation between PDAC cell lines. PCA also validated that the reduced cell feature set is sufficient to capture the morphological heterogeneity in the PDAC lines. Notably, MIA PaCa2 cells scored low on several cell features due to these cells having particularly small cell area, distorted nuclei (low nucleus roundness), and low actin abundance. Moreover, in MIA Paca2 cells, actin is localised in areas of the cytoplasm distal to the membrane. In contrast, PANC1 cells have a notably high cell and nucleus roundness, as well as high membrane/cyt actin, indicating that actin in PANC1 cells is localised predominantly at the cortex.

Distributions of nuclear RELA in the five PDAC cell lines by immunofluorescence revealed highly heterogeneous RELA responses within and across the PDAC cell lines (*Figure 2A*). To identify features that predict RELA localisation differences, we collated and incorporated normalised single-cell measurements across all PDAC cell lines and TNFα treatments into Bayesian networks, harnessing the observed variation in RELA (*Figure 2B*). Bayesian network models apply statistical inference to heterogeneous experimental data to predict the conditional dependence of components on each

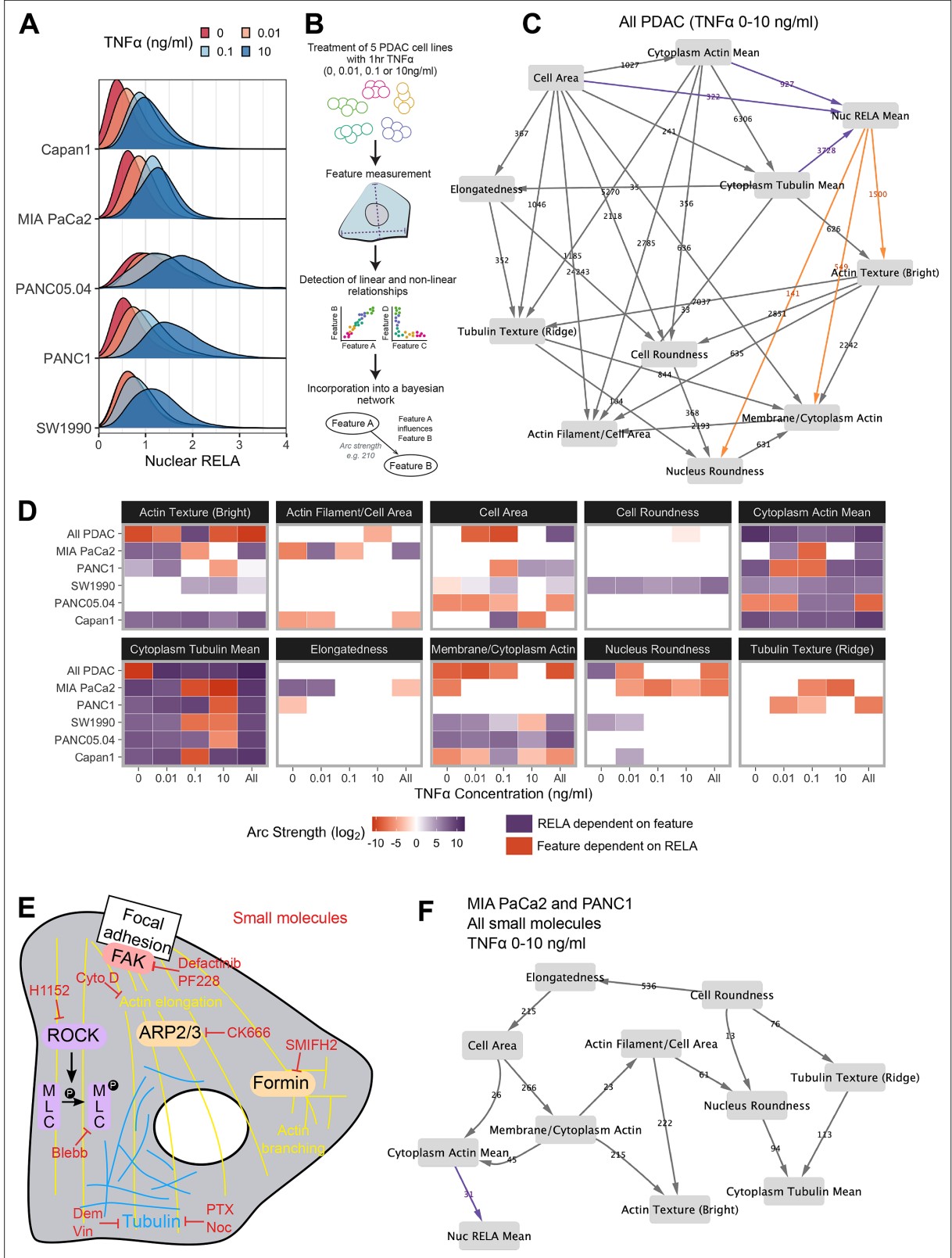

**Figure 2.** Bayesian network analysis predicts statistical dependence of RELA on actin. (**A**) Single-cell nuclear RELA intensity distributions by immunofluorescence and automated image analysis. (**B**) Schematic depicting generation of a Bayesian network model for probabilistic relationships between cell features in PDAC cells. PDAC cells were treated with 0, 0.01, 0.1, or 10 ng/ml TNFα for 1 hr. Using automated image analysis of cell markers, features are measured on a single-cell basis and Bayesian analysis is used to detect linear and non-linear relationships between the features.

*Figure 2 continued on next page*

*Figure 2 continued*

These relationships are incorporated into a Bayesian network model, which is an influence diagram consisting of nodes, each representing a measured feature, and arcs between the nodes that depict predicted dependencies between the nodes. Network was generated using single-cell data with 1000 single cells sampled per cell line (MIA PaCa2, PANC1, Capan1, SW1990, and PANC05.04), TNFα dose and biological repeat (n = 3 biological repeats, each with four wells/technical replicates). (**C**) Bayesian network model incorporating data from all PDAC lines treated for 1 hr with 0 (solvent control), 0.01, 0.1, or 10 ng/ml TNFα. Values next to arcs represent the strength of the probabilistic relationship expressed by the arc (arc strength). Orange arcs connect features predicted to depend on nuclear RELA mean, and purple arcs connect features predicted to influence nuclear RELA mean. (**D**) Dependencies involving nuclear RELA mean in Bayesian network models generated with single-cell data for individual treatments or cell lines, or for all cell lines collated (top row in each cell feature section), or all treatments collated (rightmost column in each cell feature section). Purple indicates that nuclear RELA mean is predicted to depend on the cell feature in the Bayesian network model. Orange represents that a cell feature is predicted to depend on nuclear RELA intensity. Dependency strengths are calculated as $log_2(|arc\ strength|)$, multiplied by –1 for dependencies of cell features on nuclear RELA intensity. (**E**) Schematic indicating small molecules targeting the cytoskeleton. CK666 inhibits the ARP2/3 complex that mediates actin filament nucleation and branching (*Mullins et al., 1998*). SMIFH2 inhibits formins (*Rizvi et al., 2009*), which produce long straight filaments by promoting actin nucleation and filament elongation (*Pruyne et al., 2002*). Cytochalasin D binds to the growing end of actin filaments and inhibits polymerisation (*Schliwa, 1982*). H1152 targets Rho-kinase (ROCK), preventing ROCK phosphorylation of myosin light chain that normally promotes actin-binding and contractility, while blebbistatin blocks myosin II ATPase and actin contractility (*Sasaki et al., 2002*; *Kovács et al., 2004*). Tubulin-targeting drugs prevent MT assembly (vinblastine and nocodazole), limit MT formation and cause MT depolymerisation (demecolcine), or stabilise MTs and prevent disassembly (paclitaxel) (*Spencer and Faulds, 1994*; *Vasquez et al., 1997*; *Gigant et al., 2005*). Focal adhesion kinase (FAK) regulates turnover of focal adhesions, which are integrin-containing complexes linking intracellular actin to extracellular substrates. (**F**) Bayesian network model generated by single-cell data from MIA PaCa2 and PANC1 cells treated separately with the small molecules in (**E**) for 2 hr, then simultaneously treated with TNFα (0, 0.01, 0.1, or 10 ng/ml) for 1 hr. Numbers indicate arc strengths. In the presence of small molecule inhibition of the cytoskeleton, nuclear RELA mean is predicted to be dependent on cytoplasm actin mean alone, indicated by the purple arc connecting 'cytoplasm actin mean' and 'nuc RELA mean'. Cells were analysed from three biological repeats, each with four wells/technical replicates. Numbers of cells per treatment and cell line are included in *Figure 2—source data 1* (range 9,800–20,530 cells per treatment/cell line).

The online version of this article includes the following source data and figure supplement(s) for figure 2:

**Source data 1.** Excel file with raw data and Bayesian arc strengths for *Figure 2* and *Figure 2—figure supplements 1–3*.

**Figure supplement 1.** Automated image analysis of RELA localisation, cell shape, and the cytoskeleton in human cells.

**Figure supplement 2.** Principal component analysis (PCA) of five human PDAC cell lines with a reduced set of cytoskeletal and cell shape features.

**Figure supplement 3.** Dose–responses for cytoskeleton-targeting drugs.

---

other. Bayesian network models appear as influence diagrams consisting of nodes, each representing a measured feature, and arcs that depict predicted dependencies between the nodes. These dependencies represent linear and non-linear relationships, direct and indirect interactions, and illustrate multiple interacting nodes simultaneously (*Sachs et al., 2005*). We employed a hybrid class of Bayesian algorithm ('rsmax2') that generates models with unidirectional arcs using a combination of constraint-based and score-based approaches (*Scutari et al., 2018*).

In order to provide the greatest heterogeneity in nuclear RELA and cytoskeletal/cell shape features for Bayesian model generation, we collated data from all five PDAC lines and TNFα doses (0, 0.01, 0.1, and 10 ng/ml), shown in *Figure 2C*. This model indicated that nuclear RELA measurements are correlated to and predicted to be dependent on cytoplasmic actin and tubulin intensity, suggesting that cytoskeletal dynamics influence heterogeneity in nuclear RELA translocation with TNFα. Nuclear RELA is also predicted to be dependent on cell area, although with a lower strength of the probabilistic relationship compared to actin/tubulin, while nucleus roundness is predicted to be dependent on nuclear RELA. Interestingly, both actin texture and the ratio of membrane to cytoplasm actin are also predicted to be dependent on RELA, suggesting that RELA and actin dynamics are interdependent. Altogether, our data suggest that the influence of cell shape on RELA translocation we have previously described in breast cancer cells (*Sero et al., 2015*; *Sailem and Bakal, 2017*) is likely mediated through cytoskeletal changes.

To understand how inter-line differences in cytoskeletal organisation may influence RELA translocation dynamics, we additionally generated Bayesian network models by subsetting data by cell line and TNFα concentration. We summarised dependencies involving nuclear RELA in *Figure 2D*. In contrast to our prior findings using Bayesian modelling of RELA and cell shape with breast cancer cells in the absence of cytoskeletal measurements (*Sero et al., 2015*; *Sailem and Bakal, 2017*), here we found a general lack of dependence of nuclear RELA on cell shape features, but identified strong and consistent dependencies of nuclear RELA on cytoplasm actin and tubulin, as well as actin texture,

in several PDAC cell lines and TNFα doses (*Figure 2D*). These data computationally predict that actin and tubulin abundance and actin distribution within the cell influence RELA nuclear translocation.

When using datasets of sufficient size, deriving Bayesian models based on largely stochastic and relatively small fluctuations in variables can provide insight into the influence of cytoskeletal components, shape, and RELA on each other (*Figure 2C*). But molecular interventions provide a means to further drive ordering of connections by identifying regions in state where normally correlated variables become conditionally independent (*Pe'er et al., 2001*; *Sachs et al., 2005*). Consequently, we created an additional dataset where cells were imaged following perturbations of tubulin, actin, myosin, or focal adhesion (FA) dynamics (*Figure 2E*). Such perturbations are intended to alter the value of variables/features (i.e. measures of actin organisation) beyond those observed in normal populations.

For example, CK666 inhibits the ARP2/3 complex – a key mediator of actin filament nucleation and branching (*Mullins et al., 1998*), while SMIFH2 inhibits formins (*Rizvi et al., 2009*), which promote actin nucleation and elongation of pre-existing filaments to produce long straight filaments (*Pruyne et al., 2002*). H1152 targets Rho-kinase (ROCK), preventing ROCK phosphorylation of myosin light chain that normally promotes actin-binding and consequently contractility, while blebbistatin blocks myosin II ATPase and subsequently interferes with actin contractility (*Sasaki et al., 2002*; *Kovács et al., 2004*). Tubulin-targeting drugs are commonly used in cancer chemotherapy and either prevent MT assembly (vinblastine and nocodazole), limit MT formation and cause MT depolymerisation (demecolcine), or stabilise MTs and prevent disassembly (paclitaxel) (*Spencer and Faulds, 1994*; *Vasquez et al., 1997*; *Gigant et al., 2005*; *Tangutur et al., 2017*). We ascertained optimal doses by treating MIA PaCa2 cells with dose ranges for 24 hr, or 3 hr for SMIFH2 (*Figure 2—figure supplement 3*).

We generated a single-cell dataset for use in Bayesian modelling by treating MIA PaCa2 and PANC1 cells with selected drug doses for 2 hr, then simultaneously with TNFα (0, 0.01, 0.1, and 10 ng/ml) for 1 hr and input these data into the same Bayesian algorithm as above (rsmax2) (*Figure 2F*). Interestingly, the Bayesian network following perturbations revealed that 5/6 of the variables observed to correlate with RELA (whether they influenced RELA or were influenced by RELA) in untreated cells were independent of RELA in the drug-treated network. Only 'cytoplasmic actin mean' remained as an influencing variable following drug-treatment, suggesting that actin abundance is a key regulator of RELA nuclear localisation. While other variables (i.e. tubulin and cell shape) can indirectly influence RELA, they do so via regulating actin network organisation.

## TNFα dose and duration determine profiles of RELA-dependent gene expression in PDAC cells

As RELA is known to be involved in feedback loops with transcriptional targets, with the best-studied example being IκBα (*Brown et al., 1993*; *Scott et al., 1993*; *Sun et al., 1993*), we sought to identify the transcriptional targets of RELA in PDAC and whether any are known regulators of actin.

To this end, we carried out RNA-seq analysis of PDAC cells at an early (1 hr) and late (5 hr) time-point with varying TNFα doses (0.01, 0.1, or 10 ng/ml TNFα), using MIA PaCa2 and PANC1 cells expressing endogenously tagged RELA-GFP and doxycycline (dox)-inducible IκB super repressor (IκB-SR). IκB-SR induction was used to determine which genes require RELA nuclear translocation. A total of 254 genes were differentially expressed in high TNFα (0.1 or 10 ng/ml) versus basal conditions, with the majority of genes (186) upregulated by TNFα (*Figure 3A and B*).

To identify patterns of gene expression across TNFα doses and durations, the 254 TNFα-regulated genes were organised by hierarchical clustering into seven clusters, each with distinct expression dynamics or dependence on RELA (*Figure 3C and D*). Cluster 3 shows genes with clear dependence on RELA activation in both MIA PaCa2 and PANC1 cells and has genes significantly upregulated by TNFα in a dose-dependent manner. Moreover, cluster 3 genes are more highly upregulated by TNFα in MIA PaCa2 compared to PANC1 cells, which is consistent with the RELA dynamics observed by live imaging whereby 0.1 ng/ml is sufficient to increase RELA nuclear localisation in MIA PaCa2 but not PANC1 cells (*Figure 1D*). Cluster 3 genes also have higher RELA expression at 5 hr TNFα compared to 1 hr TNFα in both cell lines, indicating that they are associated with prolonged RELA activation. As expected from the RELA and TNFα dose-dependent properties of this cluster, known RELA targets such as RELB, NFKB2, and BIRC3 are present (*Lombardi et al., 1995*; *Bren et al., 2001*; *Frasor et al., 2009*), and also have the immune ligands CD70 and CD83. Notably, cluster 3 contains Rho

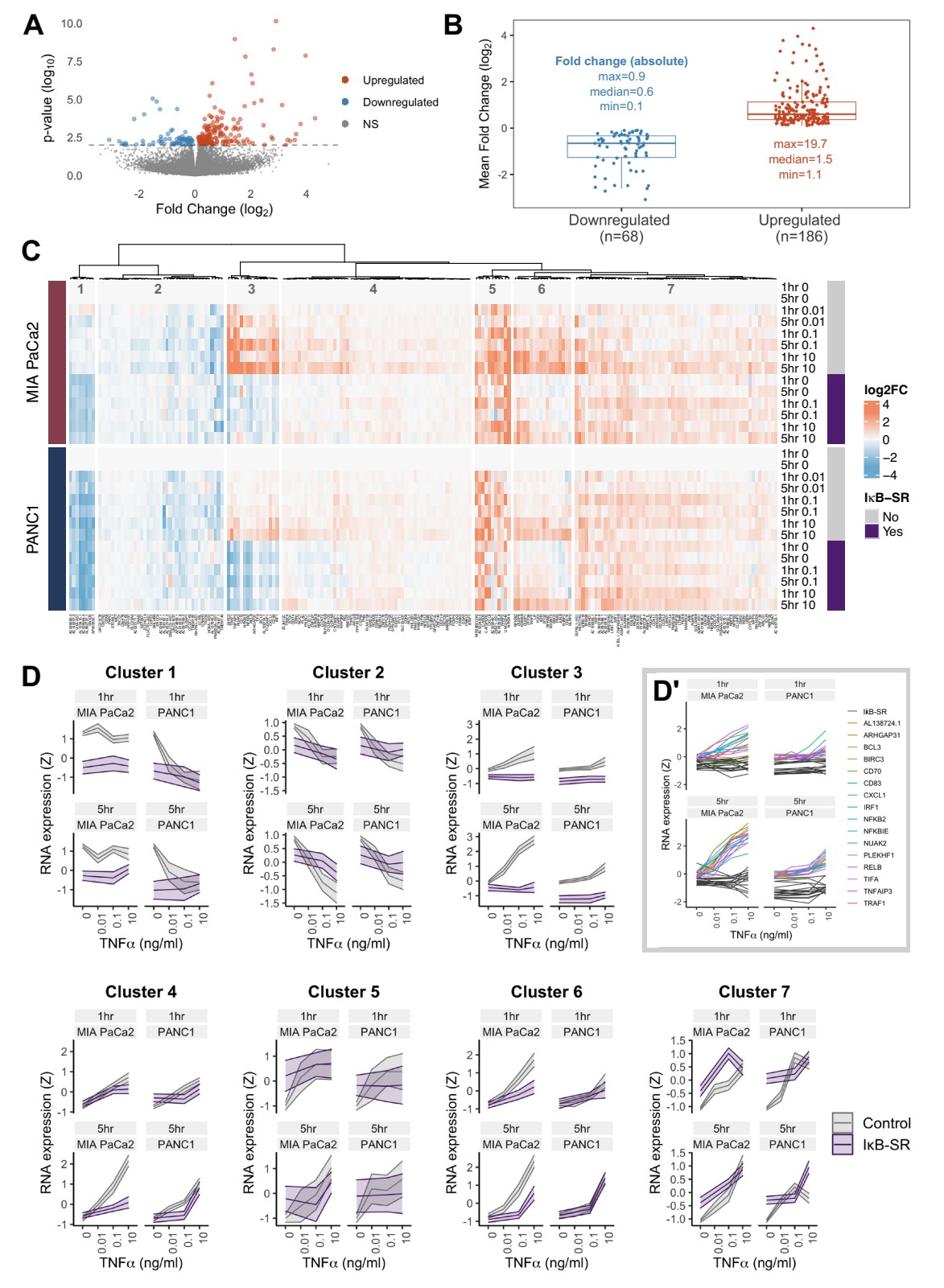

**Figure 3.** RNA-seq analysis of genes regulated by TNFα and RELA in human PDAC cells. (**A**) Volcano plot of p-value against mean fold-change (log$_2$) per gene comparing RNA expression with high TNFα (0.1 or 10 ng/ml) to control conditions (no TNFα) across MIA PaCa2 and PANC1 cells, using abundance at both 1 and 5 hr (handled as pseudo-replicates). Counts were normalised and log$_2$ fold-changes were calculated using DESeq2. Two technical replicates were processed per cell line and treatment. (**B**) Mean fold-change (log$_2$) across MIA PaCa2 and PANC1 cells for genes significantly

*Figure 3 continued on next page*

*Figure 3 continued*

downregulated (n = 68) or upregulated (n = 186) by TNFα (p<0.01) from (**A**). Also displayed are the maximum, median and minimum absolute fold-changes for downregulated and upregulated genes. (**C**) Clustered heatmap of TNFα regulated genes. Normalised counts from DESeq2 were log$_2$ transformed and relative to the respective control (no TNFα or IκB-SR) per timepoint (1 or 5 hr) and cell line (MIA PaCa2 or PANC1), then z-scored across all samples independent per gene. Columns are annotated by gene and rows are annotated by cell line, TNFα dose and time, and presence of IκB-SR. (**D**) Z-scores for all genes per cluster, faceted by cell line and treatment time (1 or 5 hr). Ribbons show the 95% confidence interval and the middle line depicts the mean. Colour corresponds to the presence of IκB-SR. (**D′**) Individual genes within cluster 3, coloured by gene for control (no IκB-SR) data and grey for all data with IκB-SR (all genes).

The online version of this article includes the following source data and figure supplement(s) for figure 3:

**Source data 1.** Excel file with bulk RNA-seq data and protein abundance from mass-spectrometry following co-immunoprecipitation of RELA-GFP.

**Figure supplement 1.** Hierarchical clustering of TNFα/RELA-regulated gene expression in PDAC cells.

GTPase-activating protein 31 (ARHGAP31) and NUAK family kinase 2 (NUAK2), which are involved in the regulation of actin (*Tcherkezian et al., 2006*; *Vallenius et al., 2011*).

Two clusters of TNFα-regulated genes in PDAC cells are downregulated by TNFα (*Figure 3C and D*) in a dose-independent manner: clusters 1 and 2. Cluster 1 consists of long non-coding RNAs (lncRNAs) and is further downregulated by IκB-SR in the MIA PaCa2 cell line only, suggesting that RELA and TNFα have antagonistic effects on the expression of these lncRNAs. In contrast, cluster 2 genes are entirely independent of RELA.

The remaining clusters (*Figure 3C and D*) contain genes upregulated by TNFα. Cluster 4 genes are similar to cluster 3 in that their expression is higher at 5 hr versus 1 hr, but cluster 4 genes have more moderate upregulation by TNFα. Moreover, cluster 4 genes appear only to be RELA-dependent with 5 hr TNFα. Clusters 5 and 7 are upregulated by TNFα in a dose-independent manner. Finally, genes in cluster 6 are dependent on TNFα dose but not duration.

Overall, we identified gene expression patterns linked to TNFα dose and duration, as well as variable dependence on RELA, suggesting that TNFα/RELA dynamics determine transcriptional output in PDAC cells.

## RELA modulates the expression of and physically interacts with non-canonical NF-κB and the IκB proteins in PDAC cells

Having observed sustained and non-oscillatory nuclear RELA with TNFα (*Figure 1*), we considered in more detail how the expression of NF-κB transcription factors and IκB family proteins is affected by TNFα dose and RELA inactivation by IκB-SR in PDAC cells, given that IκB proteins are known regulators of RELA across cell types (*Baeuerle and Baltimore, 1988*).

RNA-seq in MIA PaCa2 and PANC1 cells revealed that expression of *RELA* itself did not significantly scale with TNFα dose, while *RELB* and the transcriptionally incompetent NF-κB proteins *NFKB1* and *NFKB2* showed increasing expression with increasing TNFα (left graph in *Figure 3—figure supplement 1A*). TNFα addition significantly altered the expression of a subset of IκB protein-encoding genes, which display distinct expression dynamics: *NFKBIA* (*Figure 3*: cluster 5), *NFKBIB* (*Figure 3*: cluster 4), *NFKBIE* (*Figure 3*: cluster 3), *NFKBID* (*Figure 3*: cluster 6), and *NFKBIZ* (*Figure 3*: cluster 6). Of these, *NFKBIA* has the highest absolute RNA expression in MIA PaCa2 cells and *NFKBIB* has the highest abundance in PANC1 cells, while *NFKBID* and *NFKBIZ* have the lowest RNA expression in both cell lines (right graph in *Figure 3—figure supplement 1A*).

We compared RNA expression in TNFα-treated cells between cells with or without IκB-SR induction (Wilcoxon tests with multiple test correction), which revealed that *RELB*, *REL*, *NFKB2*, and *NFKBIE* have reduced expression with IκB-SR induction in both MIA PaCa2 and PANC1 cells (*Figure 3—figure supplement 1A′*). These findings indicate that the expression of these genes relies on nuclear translocation of RELA, while the other NF-κB and IκB genes, including the canonical RELA binding partner *NFKB1*, do not require RELA activation for expression.

We also considered how the expression of NF-κB and IκB genes is affected by the treatment duration of TNFα (*Figure 3—figure supplement 1A″*), combining data from 0.01, 0.1, and 10 ng/ml TNFα. In both MIA PaCa2 and PANC1 cells, the expression of NF-κB or IκB genes did not show any statistically significant differences between the 1 and 5 hr TNFα treatments. These results are in line with previous findings that NF-κB and IκB genes are 'early' genes upregulated rapidly and consistently following TNFα treatment (*Tian et al., 2005*; *Tay et al., 2010*).

Lastly, we checked which NF-κB/IκB proteins interact with RELA in PDAC cells. GFP-Trap followed by mass spectrometry with MIA PaCa2 cells expressing endogenously tagged RELA-GFP pulled down six NF-κB/IκB proteins (*Figure 3—figure supplement 1B*). IκBα/NFKBIA, IκBβ/NFKBIB, and IκBε/NFKBIE were pulled down at lower abundance with increasing TNFα dose, in line with the well-studied degradation of IκB proteins downstream of TNFα stimulation (*Chen et al., 1995*). Interestingly, the NF-κB protein REL also had reduced interaction with RELA with TNFα. In contrast, NFKB1 and NFKB2 showed increased interaction with RELA with TNFα. RELA therefore appears to form NF-κB heterodimers with REL, NFKB1 and NFKB2 in PDAC cells.

## The NF-κB signalling components IκBβ and RELB and the actin modulators ARHGAP31 and NUAK2 regulate TNFα-stimulated RELA nuclear localisation in PDAC cells

To test for potential feedback loops regulating RELA in PDAC, we evaluated the effect of siRNA knockdown of TNFα-regulated genes on TNFα-stimulated RELA dynamics. We targeted the genes encoding the known RELA inhibitors IκBα, IκBβ, and IκBε, which were pulled down with RELA-GFP by GFP-Trap, and the non-canonical NF-κB transcription factors RELB and NFKB2. In addition, we tested knockdown of cluster 3 (from *Figure 3*) genes *NUAK2* and *ARHGAP31*, which are known actin modulators (*Tcherkezian et al., 2006*; *Vallenius et al., 2011*). MIA PaCa2 and PANC1 cells expressing RELA-GFP were transfected with siRNAs for 48 hr, then live-imaged for −2 hr to +12 hr relative to the addition of 10 ng/ml TNFα or solvent control (*Figure 4A–C*). We used non-targeting (NT) siRNA as a negative control for comparison and verified transfection efficacy using RELA siRNA, which abrogated the nuclear RELA signal ('RELA' panels in *Figure 4A*), in addition to qRT-PCR (*Figure 4—figure supplement 1*), which showed on-target reduction in RNA expression by each siRNA.

We considered whether each siRNA affected total nuclear RELA occupancy by quantifying the area under the curve (AUC) of nuclear RELA signal (*Figure 4B*) for 12 hr control or TNFα treatment. We also quantified the fold-change in RELA AUC (TNFα/control), which informs how each gene modulates the response of RELA to TNFα. As expected, *RELA* siRNA demonstrated the most significant reduction in the AUC of nuclear RELA signal compared to the non-targeting siRNA in both cell lines in both basal conditions and with TNFα. In both cell lines, depletion of *NFKBIB* also caused significant upregulation RELA responsiveness (fold-change) to TNFα, while PANC1 cells also displayed RELA upregulation with *NFKBIA* or *ARHGAP31* depletion and RELA downregulation with *NFKB2* depletion. These results suggest that IκBβ plays a significant role in suppressing TNFα induction of nuclear RELA in PDAC cells. Interestingly, nuclear RELA AUC in control conditions was affected by all of the tested NF-κB pathway components in PANC1 cells, indicating that RELA is highly modulated by NF-κB/IκB proteins in the absence of stress or inflammatory cues.

We also considered the effects of siRNA gene depletion on acute (1 hr TNFα) and sustained (12 hr TNFα) nuclear RELA abundance in PDAC cells (*Figure 4—figure supplement 2A and B*). Of note, Wilcoxon tests with multiple test correction showed that *ARHGAP31* regulates both the 1 hr and 12 hr levels of nuclear RELA fold-change (TNFα/control) in both MIA PaCa2 and PANC1 cells. In general, *ARHGAP31* depletion upregulated nuclear RELA localisation in response to TNFα, suggesting that ARHGAP31 suppresses inflammatory activation of RELA. *NUAK2* depletion upregulates nuclear RELA following 1 hr TNFα in PANC1 cells, but downregulates nuclear RELA following 12 hr TNFα in MIA PaCa2 cells. These results suggest that the dynamics between RELA with NUAK2 are complex as they are both cell line and time-dependent. In terms of knockdown of NF-κB protein encoding genes, depletion of the non-canonical gene *RELB* significantly affected the RELA response to TNFα in the MIA PaCa2 cell line, with *RELB* depletion downregulating nuclear RELA (fold-change) at 1 hr and upregulating nuclear RELA (fold-change) at 12 hr, suggesting that RELB regulates both the 'early' and 'late' nuclear translocation responses of RELA to TNFα in MIA PaCa2 cells. While no NF-κB pathway components appear to regulate the fold-change of nuclear RELA in PANC1 cells to 1 hr TNFα, the knockdown of several NF-κB/IκB genes upregulated nuclear RELA in PANC1 cells with 12 hr treatment: *NFKBIA*, *NFKBIB*, *NFKBIE*, and *RELB*. Finally, the fold-change in nuclear RELA signal with 12 hr TNFα was significantly upregulated by *NFKBIB* depletion in PANC1 cells, suggesting that IκBβ plays a role specifically in the 'late' response of RELA to TNFα in this cell line.

Taken together, we characterised the NF-κB pathway in PDAC cells and additionally identified that the known actin modulators *ARHGAP31*/*NUAK2* are both RELA targets (*Figure 3*) and regulators

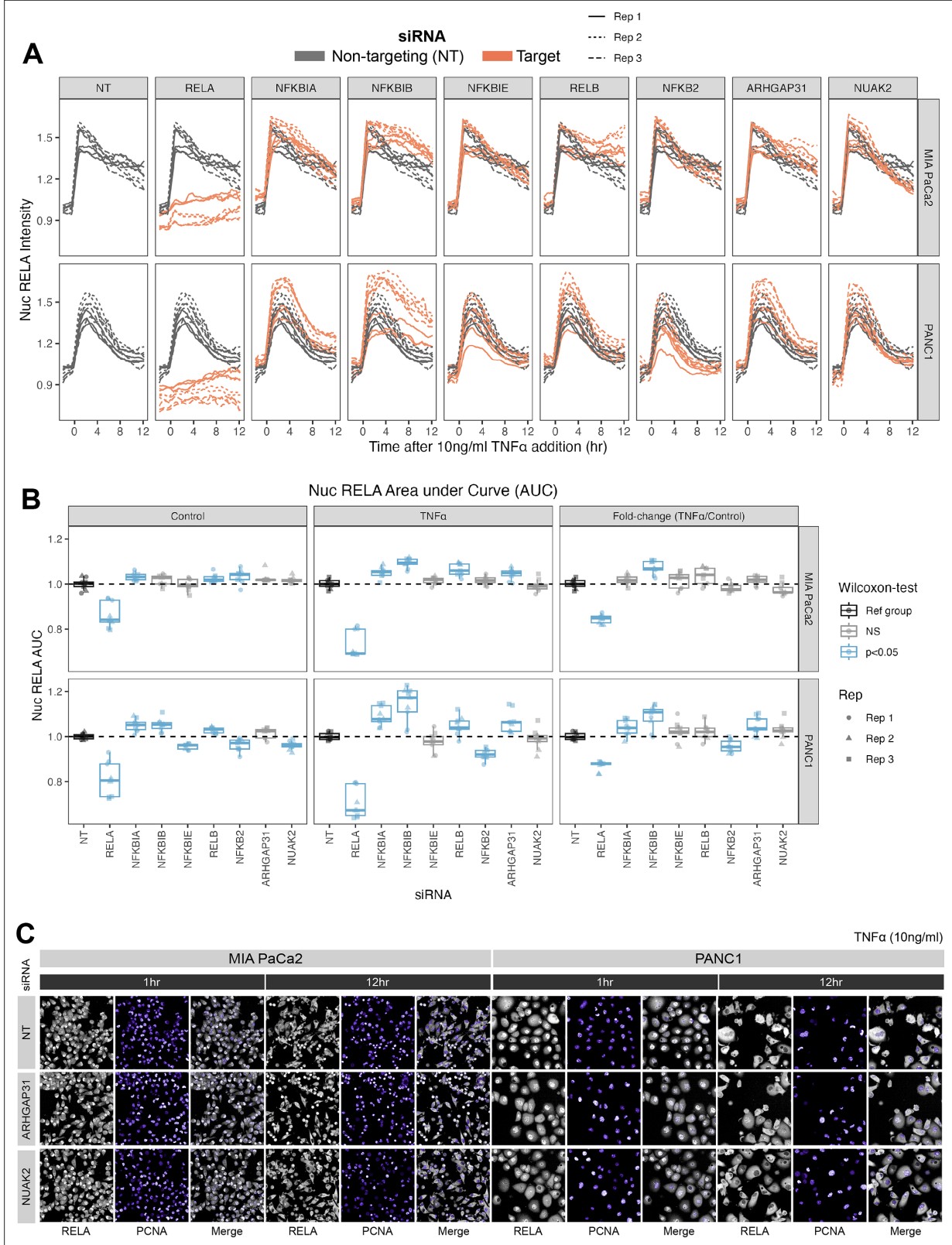

**Figure 4.** Effects of siRNA knockdown of genes involved in TNFα and actin signalling on RELA dynamics. (**A**) Tracks of RELA-GFP intensity measurements relative to the timing of 10 ng/ml TNFα addition (–2 hr to +12 hr) following 48 hr siRNA incubation, across two PDAC cell lines: MIA PaCa2 and PANC1. Each panel corresponds to an siRNA, with the three biological/experimental replicates shown in different line styles (solid, dotted, and dashed). Each line depicts well mean data (n = 3 wells/technical replicates per biological repeat). Grey lines represent RELA-GFP measurements

*Figure 4 continued on next page*

*Figure 4 continued*

with non-targeting (NT) siRNA and are identical in all facets. Orange lines show RELA-GFP tracks from cells with siRNA depletion for the target gene indicated in the column facet. (**B**) Area under the curve (AUC) measurements (see 'Materials and methods') for nuclear RELA intensity over time for each siRNA gene knockdown, for MIA PaCa2 and PANC2 cells treated with 10 ng/ml TNFα or control treatment. All data are normalised by dividing the nuclear RELA AUC by the mean AUC for the NT siRNA control by biological repeat, cell line and treatment (TNFα versus control). Statistical results are shown for Wilcoxon tests with Benjamini–Hochberg correction for multiple comparisons, testing for differences in AUC measurements between technical replicates for each siRNA to the non-targeting siRNA control, independently by cell line and treatment (TNFα versus control). Blue boxplots represent statistically significant (p<0.05) results and grey boxplots show statistically non-significant results. n = 3 experimental repeats, each with three wells/technical replicates per treatment (siRNA, TNFα dose, and cell line). Experimental repeats are depicted by shape (Rep 1 = circle, Rep 2 = triangle, Rep 3 = square). AUC fold-changes (rightmost facet) are calculated by dividing each measurement by the mean of the control (no TNFα) measurements for each siRNA. (**C**) Snapshots from confocal microscopy timelapse imaging of endogenous RELA-GFP (grey) and PCNA-Scarlet (nuclear marker, shown in purple) in MIA PaCa2 and PANC1 cells following NT, ARHGAP31, or NUAK2 48 hr siRNA treatment, followed by 1 or 12 hr 10 ng/ml TNFα addition.

The online version of this article includes the following source data and figure supplement(s) for figure 4:

**Source data 1.** Excel file with nuclear RELA intensity measurements from live imaging analysis with siRNA gene knockdown.

**Figure supplement 1.** Validation of siRNA knockdown by qRT-PCR.

**Figure supplement 2.** Mean nuclear RELA intensity in MIA PaCa2 and PANC1 cells following siRNA and TNFα treatment.

of nuclear RELA with TNFα (*Figure 4*), providing a potential mechanism mediating the relationship between actin and RELA.

## Discussion

To characterise previously unknown single-cell RELA dynamics in PDAC cells, we used CRISPR-CAS9 genome editing to tag RELA endogenously with GFP in the human PDAC cell lines MIA PaCa2 and PANC1. Using live imaging and automated image analysis, we characterised RELA responses to the cytokine TNFα, which is upregulated with PDAC progression (*Zhao et al., 2016*). Strikingly, PDAC cells show atypical RELA dynamics compared to reports in cells from other tissues (*Ashall et al., 2009*; *Tay et al., 2010*; *Sero et al., 2015*), as we observed that PDAC cells maintain prolonged nuclear RELA localisation and RELA responses are cell cycle independent. However, we identified that a key difference between MIA PaCa2 and PANC1 cells is that MIA PaCa2 cells respond with higher and sustained RELA nuclear localisation compared to PANC1 cells, while PANC1 cells have a damped RELA response over time. Similar to other cell lines (*Ashall et al., 2009*; *Tay et al., 2010*; *Sero et al., 2015*), we found that RELA nuclear translocation in PDAC cells occurs immediately following TNFα addition and peak nuclear RELA localisation occurs around 1 hr post-TNFα.

The distinct RELA dynamics between PDAC cells with reports from other tissues may be attributed to the use of knock-in RELA-GFP in this study since many studies used exogenous RELA fusion constructs (*Ashall et al., 2009*; *Tay et al., 2010*; *Sero et al., 2015*). As RELA upregulates several genes involved in positive and negative feedback (*Collart et al., 1990*; *Libermann and Baltimore, 1990*; *Brown et al., 1993*; *Scott et al., 1993*; *Sun et al., 1993*), which we observed in the present study using RNA-seq, RELA overexpression could interfere with its intrinsic dynamics. Nonetheless, other studies that have tagged RELA endogenously did detect oscillations, including MEFs (*Sung et al., 2009*; *Zambrano et al., 2016*) and MCF7 breast cancer cells (*Stewart-Ornstein and Lahav, 2016*).

Here, we identified that MIA PaCa2 and PANC1 cells in all cell cycle stages are responsive to 0.1 and 10 ng/ml TNFα, with minimal dissimilarities in RELA translocation responses between cell cycle stages in terms of mean nuclear RELA or the amplitude and timing of peak RELA. In contrast, RELA translocation responses to TNFα were identified in HeLa cells by *Ankers et al., 2016* as dependent on the cell cycle phase at the time of TNFα addition in a study using double thymidine block to synchronise cells, or Fluorescent Ubiquitination-based Cell Cycle Indicator (FUCCI) labelling to infer cell cycle phase. HeLa cells treated with 10 ng/ml TNFα in late G1 displayed a stronger response than the population average, while S-phase cells showed a suppressed or delayed response. Furthermore, RELA was found to interact with E2F1, a transcription factor regulating the G1 to S transition, in late G1 when E2F1 levels are highest during the cell cycle (*Ankers et al., 2016*). Conflicting reports of whether TNFα-induced RELA translocation is cell cycle regulated may be due to cancer cell type-specific

deregulation of cell cycle proteins (*Cordon-Cardo, 1995*; *Otto and Sicinski, 2017*) or RELA may not physically interact with E2F1 in PDAC cells in general or specifically in unsynchronised cells.

The high sensitivity of PDAC cells to TNFα and lack of oscillations suggest that PDAC cells may suppress negative feedback imposed by IκBα, as *Hoffmann et al., 2002* demonstrated that oscillations in NF-κB DNA binding are due to negative feedback with IκBα, which is encoded by a gene (*NFKBIA*) upregulated by NF-κB factors. Hoffman et al. also identified that cells with high IκBβ and IκBε expression, or absence of IκBα, lose oscillations stimulated by TNFα. These findings motivated our inspection into RELA interaction with IκB proteins and upregulation of IκB genes by TNFα in PDAC cells. Using co-immunoprecipitation of RELA with MIA PaCa2 cells, we found that RELA binds to IκBα, IκBβ, and IκBε and binding to each is suppressed in high TNFα concentrations. However, contrary to our expectations based on the findings by Hoffmann et al., we did not observe RELA oscillations with *NFKBIB* or *NFKBIE* depletion in MIA PaCa2 or PANC1 cells, suggesting that the lack of RELA oscillations in these cell lines may be due to enhanced positive feedback and/or insufficient *NFKBIA* expression.

Co-immunoprecipitation also revealed that the NF-κB protein REL had reduced interaction with RELA with TNFα. This is in line with a recent study (*Rahman et al., 2022*) that endogenously fluorescently labelled RelA and c-Rel in double knockin mice and used fluorescence cross-correlation spectroscopy to probe for binding/dimerisation in primary ear fibroblasts to quantify the abundance of RelA:c-Rel heterodimers pre- and post-TNFα. *Rahman et al., 2022* identified that the relative abundance of the RelA:c-Rel dimer was higher in the nucleus of resting cells then higher in the cytoplasm of TNFα-stimulated cells, despite the individual concentrations of RelA and c-Rel increasing in the nucleus with TNFα. Our combined results suggest that binding of RELA and REL is reduced with TNFα in multiple cell types and in both human and mouse models.

We used Bayesian modelling as an unbiased and high dimensional approach to determine whether descriptors of cell shape and the cytoskeleton correlate with the observed heterogeneity in RELA localisation, having previously used Bayesian modelling to show that RELA localisation in breast cells is strongly dependent on neighbour contact, cell area, and protrusiveness in the presence and absence of TNFα (*Sero et al., 2015*). In the present study, we extended the analysis to include measurements of actin and tubulin organisation. Since Bayesian modelling relies on heterogeneity in measurements to make predictions, we used a dataset with five PDAC cell lines with high intra- and inter-line variability in RELA, as well as distinct cell shape and cytoskeleton features between the cell lines. We independently ran Bayesian modelling with a dataset of cells with perturbation of the cytoskeleton using small molecules, based on the seminal study by *Sachs et al., 2005* that used Bayesian inference to derive cell signalling networks. Consistently among our models, differences in cytoplasmic actin intensity, as well as measures of actin localisation (cortical versus cytoplasmic actin), were predictive of differences in nuclear RELA between single PDAC cells.

Interestingly, the Bayesian-inferred arrows in the RELA network generated from TNFα-treated cells (*Figure 2C*) sometimes point in opposite directions compared to the network from cells treated with biochemical inhibitors (*Figure 2F*). This arises from the inherent properties and flexibility of Bayesian networks (*Scutari et al., 2018*). These networks use a directed acyclic graph (DAG) to represent global probability distributions broken down into smaller local distributions, ensuring no loops or cycles. Multiple valid DAG configurations can exist, so the dependence between two variables A and B can be represented as either A→B or B→A if both are probabilistically equivalent (*Heckerman et al., 1995*; *Scutari, 2010*; *Scutari et al., 2019*). Thus, reversing an arc can leave the overall network structure unchanged as long as local distributions remain consistent with the data.

One route through which cytoskeletal structures may influence RELA dynamics is by inducing changes in TNF receptor turnover or conformation. In the literature, evidence exists for the modulation of nuclear RELA by actin and tubulin inhibitors (*Bourgarel-Rey et al., 2001*; *Németh et al., 2004*), in addition to TNFα/RELA regulation of the cytoskeleton (*Georgouli et al., 2019*; *Huber et al., 2004*). However, there is limited evidence for cytoskeletal regulation of TNF receptors. One example is that the actin-binding protein Filamin interacts with TNF receptor-associated factor 2 (TRAF2), which is not itself a TNF receptor but is involved in TNF receptor intracellular signal transduction (*Leonardi et al., 2000*). On a related topic, a recently published study (*Alraies et al., 2024*) subjected dendritic cells to space confinement and found upregulation of the chemokine receptor *Ccr7*, in a manner dependent on expression and function of IKKβ and the lipid metabolism enzyme cPLA$_2$. In turn, the

prostaglandin $E_2$ receptor ($PGE_2$) receptor, which is in the $cPLA_2$ pathway, induces NF-κB nuclear translocation. This study also identifies the role of ARP2/3 activity in regulating $cPLA_2$ activation via nuclear envelope tensioning. Interestingly, transcriptomics of confined $cPLA_2^{WT}$ and $cPLA_2^{KO}$ dendritic cells revealed correlated expression between *Ccr7*, the gene encoding IKKβ (*Ikbkb*), and the major subunits of ARP2/3 (*Actr2* and *Actr3*). Overall, the impact of cytoskeletal dynamics on TNF receptors remains a source for further study.

To identify potential regulatory loops involving RELA in PDAC, we used RNA-seq analysis with MIA PaCa2 and PANC1 cells treated with varying TNFα doses +/-IκB-SR. We identified 254 genes significantly regulated by TNFα, which may be viewed as candidates for targeting NF-κB signalling or output in PDAC. Identifying novel targets for PDAC is important as the 5-year survival rate of non-resected PDAC patients has remained unchanged from 1975 to 2011 (*Bengtsson et al., 2020*). Moreover, RELA is hyperactive in 50–70% of PDAC tumours (*Wang et al., 1999*; *Weichert et al., 2007*) and contributes to both cancer progression (*Fujioka et al., 2003*; *Melisi et al., 2009*) and resistance to chemotherapy (*Bold et al., 2001*; *Kunnumakkara et al., 2007*).

Of note, we present the first evidence that *ARHGAP31* is a transcriptional target of TNFα or RELA. Moreover, ARHGAP31 is not previously associated with pancreatic cancer, but has well-studied roles in actin modulation, since ARHGAP31 is the human orthologue for the mouse protein CdGAP (mCdc42 GTPase-activating protein) and ARHGAP31 inactivates the GTPases CDC42 and RAC1 (*Lamarche-Vane and Hall, 1998*; *Tcherkezian et al., 2006*). CDC42 is involved in the formation of premigratory filopodia in PDAC cells and promotes invasiveness (*Razidlo et al., 2018*), while RAC1 expression is required for the development of PDAC tumours, in addition to the formation of the ADM and PanIN precursors in KRAS$^{G12D}$ mouse models (*Heid et al., 2011*), indicating that RAC1 may play a role in actin remodelling during PDAC initiation.

Our results also demonstrate that TNFα upregulates RNA expression of the kinase-encoding gene *NUAK2* (SNARK), while *NUAK2* expression is abrogated in the presence of IκB-SR. Our identification of a feedback loop between RELA nuclear translocation and *NUAK2* expression is analogous to prior findings that NUAK2 increases the activity of the mechanosensitive transcriptional coactivator YAP through stimulation of actin polymerisation and myosin activity (*Yuan et al., 2018*). A potential source for further study could consider myosin phosphatase target subunit 1 (MYPT1), a substrate for NUAK2 (*Yamamoto et al., 2008*) that is reported to mediate NUAK2 regulation of actin stress fibres in growing cells (*Vallenius et al., 2011*). From a therapeutic perspective, NUAK2 may be a promising target to follow-up for potential use in PDAC therapy as kinases contain an ATP-binding cleft that is a druggable pocket and can contain additional druggable sites distal to the ATP or substrate binding pockets conferring inhibitor specificity (*Lamba and Ghosh, 2012*). However, NUAK2 is a prognostic marker for PDAC and is associated with a favourable prognosis according to The Human Protein Atlas, which proposes a potential tumour suppressor role for NUAK2 in PDAC, suggesting that enhancing NUAK2 may be a favourable strategy for PDAC. Therefore, our data provide an incentive for the exploration of a RELA-NUAK2 signalling axis in PDAC progression and response to therapy.

# Materials and methods

Further information and requests for resources should be directed to and will be fulfilled by Chris Bakal (chris.bakal@icr.ac.uk).

## Cell lines and cell culture

Cell lines were maintained at 37°C and 5% $CO_2$ in Dulbecco's Modified Eagle Medium (DMEM; Gibco) supplemented with 10% heat-inactivated fetal bovine serum (Sigma) and 1% penicillin/streptomycin (Gibco). MIA PaCa-2, PANC-1, Capan-1, SW-1990, and Panc05.04 were obtained from ATCC.

## Generation of cell lines with fluorescently tagged RELA and PCNA by CRISPR-CAS9

RELA and PCNA were tagged endogenously at each C-terminus using CRISPR-CAS9-mediated gene editing in MIA PaCa2 and PANC1 cells. RELA was tagged with enhanced GFP (*Zhang et al., 1996*) and PCNA was tagged with mScarlet-I (*Bindels et al., 2017*), abbreviated here as RELA-GFP

and PCNA-Scarlet, respectively. RELA-GFP was first introduced into wildtype cell lines, then PCNA-mScarlet was added to validated RELA-GFP clones.

Homology constructs were generated by extracting the region around the stop codon of each gene by PCR. The product was used as a template to amplify the left homology arm (LHA) and right homology arm (RHA) by PCR. PCRs were carried out using High-Fidelity Q5 DNA Polymerase (NEB) according to the manufacturer's protocol. The RHA contains a mutation corresponding to the gRNA protospacer adjacent motif (PAM) to prevent repeat targeting by the Cas9 nuclease. Primers used to amplify the homology arms included overlaps for (1) a DNA cassette encoding a linker protein, the fluorescent protein, and antibiotic resistance (kindly donated by Francis Barr); and (2) the pBluescript II SK (-) vector (Agilent) following EcoRV digestion. The final homology construct was generated from the four DNA oligos by Gibson assembly using the NEB Gibson Assembly Master Mix and according to the NEB protocol.

gRNA oligos were designed using CRISPR.mit.edu. Forward and reverse oligos were phosphorylated, annealed, and ligated into a BbsI-digested pX330 U6 Chimeric hSpCas9 plasmid, gifted by Feng Zhang (*Cong et al., 2013*).

Custom oligos were synthesised by Sigma-Aldrich. gRNA oligos were designed using CRISPR. mit.edu.: TGAGTCAGATCAGCTCCTAA (RELA gRNA forward), TTAGGAGCTGATCTGACTCA (RELA gRNA reverse), CGAGGATGAAGAAGGATCTT (PCNA gRNA forward), AAGATCCTTCTTCATCCTCG (PCNA gRNA reverse). The following oligos were used for genomic DNA extraction: TGGGTCAG ATGGGGTAAGAG (RELA C-terminus forward), CCAGCTTGGCAACAGATTTA (RELA C-terminus reverse), GCCCTGGAGCCTTGATATTCA (PCNA C-terminus forward), TCTCACTTGTTCCTTGAGCT CA (PCNA C-terminus reverse). The following oligos were used for homology construct generation, using the genomic DNA as a template: ACGGTATCGATAAGCTTGATTGGGTCAGATGGGGTAAGAG (RELA left arm forward), CGCCACCACCGCTCCCACCGGAGCTGATCTGACTCAGCA (RELA left arm reverse), TTCTTGACGAGTTCTTCTGAGGAGGTGACGCCTGCCCTCC (RELA right arm forward), CCGGGCTGCAGGAATTCGATCAAGGAAGTCCCAGACCAAA (RELA right arm reverse), ACGGTATC GATAAGCTTGATGAGTTTGCAGAGCTGAAATTA (PCNA left arm forward), CGCCACCACCGCTCCC ACCAGATCCTTCTTCATCCTCGA (PCNA left arm reverse), GTTATGTGTGGGAGGGCTAAGCATTCTT AAAATTCAAGAA (PCNA right arm forward), CCGGGCTGCAGGAATTCGATTCTCACTTGTTCCTTG AGCT (PCNA right arm reverse).

Cells were transfected with homology and gRNA constructs using Lipofectamine 2000 (Thermo Fisher) according to the manufacturer's protocol. Cells were expanded and selected for antibiotic resistance for 3 weeks. FP-positive cells were selected using FACS and sorted into single cells per well in 96-well plates, and clones were expanded and tested for FP presence by amplifying and sequencing the C-terminus of the RELA and PCNA genes from genomic DNA in order to confirm the presence of the linker and eGFP or mScarlet DNA.

## Lentiviral cell line generation

HEK-293T cells were transduced using Effectene (QIAGEN) with the lentiviral plasmid pTRIPZ-IκB-SR (mouse), kindly gifted by Tencho Tenev (Pascal Meier Lab), with the packaging plasmid psPAX2 (Addgene) and envelope expressing plasmid pMD2.G (Addgene).

After 48 hr, supernatant from transduced HEK-293T cells was filtered using 0.45 µm syringe filters, then immediately added to target cells.

After 72 hr, target cells were selected with puromycin for 2–4 weeks, then single-cell sorted using FACS into 96-well plates. Clones were expanded and tested for retention of RELA-GFP in the cytoplasm in the presence of 10 ng/ml TNFα when pretreated with dox for 24 hr and normal nuclear translocation of RELA-GFP without dox treatment. In all experiments involving IκB-SR, cells were treated with dox at a final concentration of 0.1 µg/ml.

## Cell seeding and treatment for fixed image analysis

Cells were seeded at a density of 1000 cells per well in 384-well plates unless otherwise specified.

For comparison of cell shape and cytoskeletal features in the five PDAC cell lines, cells were fixed 2 days after seeding, including 1 hr TNFα treatment. The experiment was carried out three times (biological replicates) in total, each with four technical replicates (wells) per condition.

## TNFα treatment

Cells were treated with human recombinant TNFα diluted in complete medium at a final concentration of 0.1 ng/ml, or 1 ng/ml and 10 ng/ml when specified. TNFα sourced from Sino Biological was diluted in water and used to treat the panel of PDAC lines for Bayesian analysis (*Figure 2*). Due to the lack of availability, TNFα was sourced from R&D Systems and diluted in 0.1% BSA/PBS for all other experiments.

## Immunofluorescence

Cells were fixed with warm formaldehyde (FA) dissolved in PBS at a final concentration of 4% for 15 min at 37°C, then washed three times with PBS. Cells were permeabilised in 0.2% TritonX-100 (Sigma-Aldrich) dissolved in PBS for 10 min and blocked in 2% BSA/PBS for 1 hr at room temperature (RT). Cells were stained with 10 µg/ml Hoechst (Sigma-Aldrich) in PBS (1:1000) for 15 min, washed three times, and left in PBS/azide before imaging.

Cells were incubated with primary antibodies for 2 hr at RT or overnight at 4°C, washed three times with PBS, and incubated with secondary antibodies for 90 min at RT.

Primary antibodies used were rabbit anti-p65/RELA NF-κB (Abcam #16502; 1:500), rat anti-α-tubulin (Bio-Rad #MCA78G; 1:1000), and rabbit anti-pFAK Tyr397 (Invitrogen #44-624G; 1:250).

Secondary antibodies used were Alexa 647 goat anti-rat IgG (Invitrogen #A21247), Alexa 488 goat anti-rabbit IgG (Invitrogen #A11034), and Alexa 647 goat anti-rabbit IgG (Invitrogen #A21246).

For F-actin staining, cells were incubated with Alexa-568 phalloidin (Invitrogen #A12380; 1:1000) for 90 min simultaneously with secondary antibodies.

## Imaging and automated analysis of fixed cells

A minimum of 21 fields of view per well were imaged using the PerkinElmer Opera confocal microscope using a ×20 air objective. Image analysis was performed using custom image analysis scripts created and executed on PerkinElmer's Columbus 2.6.0 software platform. Scripts detected and segmented individual nuclei using Hoechst and the cytoplasm using tubulin, or RELA when tubulin is not included in the staining set. Cells touching the image border are filtered out and neighbour contact (% cell border touching another cell) for each remaining cell is calculated. The nuclear region is reduced by 1 px from the nuclear outer border from Hoechst segmentation and the ring region is set as the area 2 px to 6 px outside of the nuclear outer border. Intensities of all stains are calculated in all segmented regions on a single-cell level. Nuclear RELA measurements refer to the mean RELA signal (mean pixel intensity) in the nucleus, that is, the total RELA signal divided by the area of the nucleus.

A total of 32 geometric, cytoskeletal, and Hoechst features were measured in addition to measurements of RELA/RELA-GFP. Texture features were calculated using SER methods with region normalisation. Bright and Spot textures were smoothed to a kernel of 4 px to detect large patches (bundles) of actin/tubulin. Ridge texture was non-smoothed to detect sharp ridges (filaments) of actin/tubulin. Elongatedness was calculated as $((2 * \text{Cell Length})^2/\text{Cell Area})$. Actin filament area was measured using Columbus's 'Find Spots' function applied to the actin channel. Neighbour contact was calculated using an inbuilt Columbus algorithm calculating the percentage of a cell's border in contact with other cell borders. Grouped neighbour contact measurements were generated from non-normalised data rounded to the nearest multiple of 10.

## Live cell imaging and analysis

MIA PaCa2 RELA-GFP PCNA-Scarlet and PANC1 RELA-GFP PCNA-Scarlet cells were seeded (1000 cells/well) in a 384-well plate 1 day prior to imaging. Four fields per well were imaged at 10 min intervals with a ×20 air objective and an environmental control chamber set to 80% humidity, 5% $CO_2$, and 37°C.

For live imaging with TNFα only (no siRNA), cells were imaged for 2 hr prior to and 48 hr following TNFα addition using the Opera QEHS imaging system (PerkinElmer). Nuclear RELA and PCNA intensities were measured using Nuclitrack software (*Barr et al., 2017*), with 50–60 cells tracked per treatment, cell line, and biological replicate (n = 2). Nuclear intensity (RELA or PCNA) is calculated as the total intensity in the nucleus region divided by the nucleus area. Cells were tracked for 10 hr following TNFα treatment while the total 48 hr imaging period was used to ascertain cell fate (division or death).

Each biological replicate consisted of eight technical (well) replicates per cell line and treatment. Missing data points in the centre of tracks were imputed using the 'imputeTS' package in R.

For live imaging with TNFα and siRNA, cells were imaged for 2 hr prior to and 12 hr following TNFα addition using the Opera Phenix Plus High-Content Screening System (PerkinElmer), with two biological replicates (n = 2). Nuclear RELA was measured in all non-border cells using Harmony software (PerkinElmer). In R, data were smoothed with a Savitzky–Golay filter using the 'Signal' package (filter order $P$ = 1, filter length n = 7).

In all experiments, intensity measurements are normalised to the mean of the solvent control measurements (TNFα absence) by cell line and timepoint to account for photobleaching and laser power changes. Data with siRNAs are normalised to the non-targeting siRNA controls (solvent control).

Nuclear RELA peaks were detected in R by track following Savitzky–Golay filtering using the 'Signal' package (filter order $P$ = 3, filter length n = 11) by finding the first timepoint at which the slope of the nuclear RELA curve is ≤0.

## GFP-Trap

MIA PaCa2 RELA-GFP cells were cultured in 15 cm dishes to 80% confluence, then treated with TNFα at a final concentration of 0.01, 0.1, or 10 ng/ml, or with a solvent control. Co-immunoprecipitation (Co-IP) was carried out using GFP-Trap Agarose beads (Chromotek #gta-10) or with binding control agarose beads (Chromotek #bab-20) using RIPA lysis buffer and according to the manufacturer's protocol. Samples were analysed using mass spectrometry by the ICR proteomics core (Theodoros I Roumeliotis and Jyoti Choudhary).

## Cytoskeletal drug treatments

A total of 2000 MIA PaCa2 RELA-GFP cells/well were seeded in 384-well plates and treated the next day with the following drugs for 24 hr without TNFα at the specified dose ranges: paclitaxel (6.25–200 nM; Sigma), vinblastine (3.125–100 nM; Sigma), nocodazole (12.5–400 nM; Sigma), demecolcine (6.25–200 nM; Sigma), cytochalasin D (0.125–4 µM; Sigma), CK666 (12.5–400 µM; Sigma), H1152 (1.25–40 µM; Tocris), blebbistatin (1.25–40 µM; Sigma), PF573228 (0.625–20 µM; Tocris), and defactinib (0.625–20 µM; Selleckchem). Cells were treated with SMIFH2 (3.125–100 µM; Abcam) for only 3 hr due to reported cycles of de- and re-polymerisation of 4–8 hr and inefficacy after 16 hr (*Isogai et al., 2015*). Ranges were selected according to literature and manufacturers' recommendations. Doses for further analysis were selected based on the observed effect on the cytoskeletal target and cell morphology.

MIA PaCa2 and PANC1 RELA-GFP cells were seeded at 2000 cells/well in 384-well plates and treated the following day with selected doses of the small-molecule inhibitors for 2 hr plus additional 1 hr co-incubation with 10 ng/ml TNFα (or DMEM control) prior to fixation. n = 3 biological replicates. Cell feature measurements were calculated as fold-changes to controls (DMSO and BSA/PBS) then z-scored across TNFα treatments and cytoskeletal drugs by cell line. Nuclear RELA measurements were normalised to the mean of all measurements by cell feature across all TNFα treatments by cell line.

## RNA sequencing

A total of 300,000 MIA PaCa2 or PANC1 cells expressing endogenously tagged RELA-GFP and dox-inducible lentiviral IκB-SR (outlined above in 'Lentiviral cell line generation') were seeded in t25s to attain 30% confluence the following day. 24 hr after seeding, cells were treated with 1 µg/ml dox or DMSO. After 48 hr dox, cells were treated with TNFα at a final concentration of 0.01, 0.1, or 10 ng/ml, or with a BSA/PBS control for 1 or 5 hr. Cells were harvested using the RNeasy Plus mini kit (QIAGEN). Two technical replicates were processed per cell line and treatment.

Bulk RNA-seq was carried out using the NovaSeq 6000 (Illumina) with the NovaSeq 6000 Reagent Kit. Samples were processed and reads aligned by the ICR Genomics Facility. The genome was mapped to Hg38 release GRCh38.92. Reads were aligned using Star Aligner version 2.7.6, then processed with HTSeq 0.12.4 to provide counts. Statistical analysis was then carried out using the DESeq2 package (*Love et al., 2014*) in R. Normalised counts from DESeq2 were $\log_2$ transformed. The $\log_2$ value for the DMSO control (without dox) for the corresponding cell line (MIA PaCa2 or PANC1) and timepoint (1 or 5 hr) was negated from transformed counts. Counts were then z-scored by cell line or across

all samples as indicated. Genes with <10 counts across all samples were removed prior to statistical analysis, and genes with 0 counts for any sample were omitted from results.

## qPCR

A total of 250,000 MIA PaCa2 RELA-GFP cells were seeded per well on a 6-well plate 1 day prior to siRNA transfection by Lipofectamine RNAiMAX (Thermo Fisher). 48 hr after siRNA transfection, cells were treated with 1 hr 10 ng/ml TNFα (or BSA/PBS control), then harvested. RNA was extracted using RNAeasy (QIAGEN) and RNA to cDNA conversion was carried out using the SensiFAST cDNA synthesis kit. qRT-PCR was performed using SYBR green (Bioline) with predesigned KiCqStart SYBR Green Primers (Merck).

## Quantification and statistical analysis

To analyse cell-to-cell differences in the 35 geometric, cytoskeletal, and Hoechst features within and between PDAC cell lines, single-cell and well (mean) data were collated from all cell lines and treatments (TNFα 0, 0.01, 0.1, and 10 ng/ml) from three biological replicates. Features were normalised to the mean across all treatments and cell lines for each biological replicate. Features were reduced for Bayesian analysis by clustering normalised single-cell measurements into 10 clusters using the 'ComplexHeatmap' package in R (*Zhao et al., 2016*), clustering by the Pearson coefficient with complete linkage, as shown in *Figure 2—figure supplement 1B*. Bayesian network models and arc strengths were generated in R using normalised single-cell data for the 10 reduced features via the 'bnlearn' R package (rsmax2 method) (*Scutari, 2010*). This algorithm depicts unidirectional arcs, so reverse relationships can exist but are not as statistically likely as the directional relationships indicated. Bayesian modelling with cytoskeletal perturbation used data collated from all small molecules and TNFα doses (0, 0.01, 0.1, and 10 ng/ml) for MIA PaCa2 and PANC1 cells.

Z-scores in *Figure 2—figure supplement 2* were calculated per technical replicate using the mean and standard deviation for control measurements for each feature across all lines for each biological replicate (n = 3). Mean z-scores per feature and cell line were calculated by averaging z-scores for technical replicates across all biological replicates.

Statistical tests were carried out using the 'Rstatix' package and visualised using the 'ggpubr' package in R. PCA was carried out in R using the inbuilt 'prcomp' function using z-score data. Graphs were generated in R using the 'ggplot' package (*Wickham, 2016*). The area under the curve (*Figure 4—figure supplement 2*) was calculated in R using an 'integrate' function between the first and last imaging timepoints, with 1000 subdivisions.

## Acknowledgements

We thank Theodoros I Roumeliotis and Jyoti Choudhary for performing protein mass spectrometry and analysis. We acknowledge the ICR Genomics Facility for carrying out RNA sequencing and performing sequence alignment and counts. We are grateful to Victoria Roulstone for performing qPCR experiments (*Figure 4—figure supplement 1*). We thank Andrea Brundin for assistance with single-cell tracking. We gratefully acknowledge funding for this work by Cancer Research UK, awarded to FB (S_3567). CB was funded by a Cancer Research UK award supported by Stand Up to Cancer UK (C37275/A20146). We thank the Reviewers and eLife editorial team for their invaluable feedback and constructive comments.

## Additional information

### Funding

| Funder | Grant reference number | Author |
| --- | --- | --- |
| Cancer Research UK | S_3567 | Francesca Butera |
| Cancer Research UK (CRUK) supported by Stand Up to Cancer UK | C37275 | Chris Bakal |

| Funder | Grant reference number | Author |
|---|---|---|
| Cancer Research UK (CRUK) supported by Stand Up to Cancer UK | A20146 | Chris Bakal |

The funders had no role in study design, data collection and interpretation, or the decision to submit the work for publication.

## Author contributions

Francesca Butera, Conceptualization, Data curation, Formal analysis, Funding acquisition, Validation, Investigation, Visualization, Methodology, Writing – original draft, Writing – review and editing; Julia E Sero, Conceptualization, Supervision; Lucas G Dent, Supervision, Visualization, Writing – review and editing; Chris Bakal, Conceptualization, Supervision, Funding acquisition, Writing – original draft, Writing – review and editing

## Author ORCIDs

Francesca Butera ⓘ https://orcid.org/0000-0002-6606-4678
Julia E Sero ⓘ http://orcid.org/0000-0002-0299-9212
Lucas G Dent ⓘ https://orcid.org/0000-0001-8573-4617
Chris Bakal ⓘ https://orcid.org/0000-0002-0413-6744

## Decision letter and Author response

Decision letter https://doi.org/10.7554/eLife.86042.sa1
Author response https://doi.org/10.7554/eLife.86042.sa2

---

# Additional files

## Supplementary files

• MDAR checklist

## Data availability

All data generated for this study have been included as source data files. RNAseq data (FASTQ files and processed counts) are available from GEO under accession GSE268743.

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
