## [Editor Report]

This paper presents an important investigation of the relationship between cell morphology, actin cytoskeletal features, and NF-kappaB/RELA signaling dynamics. Solid evidence is provided using quantitative live-cell imaging of pancreatic cancer cell lines. These analyses better establish the connection of cell shape to the NF-kappaB signaling pathway, highlighting the importance of the actin network and several specific regulators in a feedback loop controlling NF-kappaB activity. Because NF-κB controls inflammation and cell survival, this study will be of interest in the fields of cancer and immune signaling.

---

## [Decision Letter]

**Decision letter after peer review:**

[Editors’ note: the authors submitted for reconsideration following the decision after peer review. What follows is the decision letter after the first round of review.]

Thank you for submitting the paper "Actin networks modulate heterogenous NF-κB dynamics in response to TNFα" for consideration by *eLife*. Your article has been reviewed by 3 peer reviewers, including John G Albeck as the Reviewing Editor and Reviewer #1, and the evaluation has been overseen by a Senior Editor. The following individuals involved in the review of your submission have agreed to reveal their identity: Myong-Hee Sung (Reviewer #3).

Comments to the Authors:

We are sorry to say that, after consultation with the reviewers, we have decided that this work will not be considered further for publication in its current form by *eLife*.

Specifically, while the reviewers found the approach and overall message of the paper to be potentially of significant interest, a number of concerns were raised about the methodology and inconsistencies within the data. After consultation, the reviewers came to the consensus that the results as they currently stand are insufficient to support the main conclusion of the paper. The main concerns are as follows:

1. The "cytoplasmic ring" method is not an appropriate approach to quantifying the nuclear/cytosolic ratio of RelA in this study. Although this approach is commonly used in the field, it is sensitive to changes in cell shape, raising the possibility that associations based on this measure could be artifacts. The reviewers have made a number of suggestions for how this issue could be mitigated and controlled.

2. The overall evidence for a causal relationship between actin morphology and NF-κB signaling is insufficient to support the main conclusion. Bayesian network analysis is useful for uncovering statistical relationships, but it cannot be used to make hard conclusions about causality. While pharmacological perturbations could in principle be used to support the associations identified by the Bayesian network analysis, the results from these experiments were inconsistent and seemingly contradictory: in some cases where the drug treatments modified the actin structures implicated by the network analysis, no clear effect on NF-κB was observed.

3. The differences in NF-κB signaling in the two PDAC cell types examined by live-cell imaging, both between the lines and between these lines and other cell types in the literature, raise significant questions that are not addressed in adequate detail to conclude that negative feedback is simply weaker in these cells. The data suggest that these cells diverge significantly from how the pathway works in other cells types. At a minimum, this difference must be presented more carefully, and it would greatly strengthen the paper if negative feedback genes could be analyzed to provide an explanation for the unusual behavior of the pathway in these cells.

While these issues prevent *eLife* from further considering this paper in its current form, we would be open to considering a new submission in which these issues are addressed, attempting to reach the same reviewers.

*Reviewer #1 (Recommendations for the authors):*

In this study, the authors set out to profile the kinetics of RELA translocation in pancreatic cancer cells. RELA translocation is a widely studied event that occurs in response to a number of inflammatory stimuli, and the authors begin by profiling RELA translocation kinetics in two PDAC cell lines. Relative to other cell lines where this analysis has been performed, the two PDAC cell lines show more sustained signaling (to different extents), rather than oscillatory nuclear localization. Like many other cell lines that have been studied, the authors observe a large degree of heterogeneity in RELA translocation within cells stimulated with physiological doses of TNF. A thorough analysis based on a knock-in PCNA marker reveals that this heterogeneity is not linked to the cell cycle. The authors then hypothesize the connection between actin morphology based on their previous work, and turn to an analysis of 5 PDAC cell lines, using a fixed-cell immunofluorescence assay to quantify RELA translocation, F-actin, and tubulin in each cell. From images of thousands of fixed cells, a series of statistical analyses are used to identify correlations between cytoskeletal morphology and RELA nuclear abundance. This approach reveals clear associations between certain cytoskeletal features and RELA translocation, with substantial differences between cell lines. A follow-up experiment correlates live-cell RELA kinetics with fixed-cell cytoskeletal stains, which provides further support for their associations. Finally, they perturb actin and tubulin function with a variety of chemical agents and quantify the resulting effects on live-cell RELA kinetics. This again shows variation between cell lines but confirms that direct perturbation of cytoskeletal processes can modulate the ability of TNF to stimulate RELA translocation. Together, these lines of evidence provide substantial support for the hypothesis that NF-κB signaling can be modulated downstream of changes in actin dynamics. While there is significant variation in how this works between cell lines, which suggests that a broader exploration of other cell lines will be needed in the future, the data shown here are, with some limitations noted below, adequate to support the authors' hypothesis.

A major strength of this manuscript is its use of quantitative imaging to exploit the existing heterogeneity between cells at a physiological dose of TNF. A series of clever analysis methods are used to draw out correlations that can be found between actin morphology (immunostained) and RELA translocation (either immunostained or by live-cell microscopy). Also important is the careful characterization of NF-κB signaling in pancreatic cancer cells, where this pathway is known to play a role. Importantly, RELA kinetics are assessed using a knock-in fluorescent protein tag, avoiding potential artifacts from overexpression. An interesting feature of this analysis is the use of Bayesian networks to model the dependencies of actin features; to my knowledge, this is a novel use of this algorithm. The drug perturbation data are also beautifully presented, in a way that reveals the full complexity of the cellular responses. In general, the data and conclusions are very clearly presented, and the authors do not shy away from describing the context-dependent intricacies of their observations. The main conclusion that NF-κB signaling is strongly affected by actin dynamics ties together two major signaling axes in cancer cells, and would be of interest to the many researchers working in these two areas. The methods used to show associations at the single-cell level will also have a significant impact on others working with mixtures of live and fixed single-cell data.

There are several weaknesses that limit the interpretability of the findings:

1. It is not clear how the effects of cell shape changes on the recorded nuclear/cytoplasmic ratio are controlled for. Cell shape is a major variable in this study and can alter the apparent (recorded) intensity of nuclear and cytoplasmic stains in many imaging and analysis pipelines, independent of any actual changes in a marker's localization. Without such an assessment, it isn't clear whether differences in RELA behavior (the main observation in the paper) are influenced by how the detection process treats cells with different morphologies.

2. It is not clear whether the cytoskeleton-linked effects are specific to RELA, or rather reflect more widespread changes in nuclear import/export rates. There are reports that changes in actin function can alter nuclear-cytoplasmic translocation (for example, PMIDs 16120220, 31444357). Understanding whether the observed actin-dependent effect on NF-κB localization is unique to this signaling pathway, or a non-specific effect on multiple pathways is important to the interpretation of the findings.

3. It is difficult to assess the importance of the observed differences in RELA translocation without additional information on the pathway's output. Translocation of RELA is an important marker and is known to modulate many downstream genes involved in inflammation. However, given the differences in RELA behavior observed here in PDAC cells, it is difficult to extrapolate from previous work to evaluate whether the actin-modulated changes impact RELA's transcriptional effects. Establishing that such differences exist would be important to conclude that actin shape changes have a biologically significant effect on NF-κB signaling with the potential to alter cellular behavior.

Suggestions that would have a major impact on the interpretability of the study:

1. NF-κB target gene expression, either at the RNA or protein level, should be assessed using a method of the authors' choice. Immunofluorescence for known NF-κB targets could be performed and coupled to the single-cell analysis, comparing expression levels between cells with different actin morphological features. Alternatively, bulk RNA or protein measurements for NF-κB targets could be made on cells treated vs. untreated with actin-perturbing agents.

2. It should also be established whether actin-mediated changes in nuclear import/export are restricted to RelA or occur more generally. Changes in overall nuclear import/export could be assessed using an NLS/NES-containing fluorescent protein reporter, followed by leptomycin B treatment, in a similar setup to the RelA translocation assay. A number of other methods would also be possible, such as testing whether other nuclear localization-based signaling reporters show differences in cells with different actin parameters.

3. The effects of cell shape on the recorded nuclear/cytoplasmic ratio should be quantified. This could be done using generic cytoplasmic and nuclear markers (ideally ones that do not translocate significantly), and testing whether their N/C ratio varies between cells of different shapes/morphological classes.

*Reviewer #2 (Recommendations for the authors):*

The work by Butera et al. investigates the relationship between the cellular cytoskeleton and NFkB dynamic response to TNF. The authors use single-cell fixed and live cells measurements of RelA dynamics and cellular morphology, pharmacological perturbations, and extensive statistical analysis to draw specific conclusions on what aspects of the actin cytoskeleton modulate TNF-dependent NFkB response. I have identified multiple issues that raise concerns that the conclusions of this paper are not supported by the data presented.

1. The possibility of confounding factors between cell morphology and the approach used by the authors to analyze NFkB dynamics (Figure 1D). The analysis of NFkB dynamics presented here is based on an image segmentation strategy that relies on a single marker, PCNA, and segmentation of the nucleus and "ring" around the nucleus. This strategy should only be applied in contexts where cell shape is not changing and is not a key factor in the investigation and it cannot be used to investigate the dependency of NFkB on cell shape. The main issue is that the "ring" is not a good proxy for the actual concentration of the fluorescent marker in the cytoplasm. For example, in a case where the marker (RelA-GFP in this case) is not changing, the measurement of the "cytoplasmic" signal can change as the cell spreads from a rectangular shape with limited lamellipodia to a cell that is very spread with thin lamellipodia. This change does not mean that the concentration of the marker in the cytoplasm changed, it just reveals the use of "ring" as a proxy is limited. This puts the key measurement of this work in doubt as it is the key theme throughout this paper. To give just one example, as the number of "neighbor contact" increases, cells tend to have thicker and less spread morphology (i.e. they are just denser in culture). Thicker cells will have more cytoplasm in the "ring" compared to very spread cells and therefore the denominator of RelA will be bigger that will cause the overall RelA ratio to be smaller which is exactly what the authors show in Figure 4D. Just to be clear, this is not just an issue with cell thickness, the estimation of total cytoplasmic intensity from a "ring" inherently depends on cell morphology. Therefore, one simply cannot state that RelA activation itself depends on cell morphology using this analysis method.

2. Bayesian network analysis does not provide a causal relationship. The authors make extensive use of Bayesian networks (BN) for inferences of dependencies (Figures 3 and 4). In this analysis, the authors aim to identify the direction of dependency between different nodes in the BN. This is problematic for two reasons. First, The direction of arrows in bayesian networks does not imply causations (i.e. modulation). The direction of arrows in BN only represents correlation and the directionality should not be interpreted as causative. Yet the authors clearly make a key distinction whether RelA depends on a morphological feature or whether the feature depends on RelA. In the literature that the authors cite (e.g. Sachs et al. 2005) pharmacological perturbation is used to infer causation, not the inference of the structure of the BN. Second, the authors did not consider that other factors, not included in the model, could influence both RelA and morphology. For example, if an unknown kinase changes both NFKB and a specific cellular feature the inferred network will still look the same. Therefore, the conclusion that actin-based cellular features modulate (i.e. cause) differences in RelA dynamics is simply unsupported by the BN analysis.

3. Lack of support by pharmacological data. Beyond correlation analysis, the authors used a pharmacological approach to show that manipulation of the cytoskeleton causes changes in RelA dynamics. The data presented in Figures 1-5 makes specific predictions. If one is to interpret the arrows in the BN as the authors present them, one will conclude that perturbations of features like "nuclear roundness" that was presented as key "modulator" of RelA dynamics will cause a change in RelA dynamics. The vast majority of the pharmacological perturbations used by the authors, even those that caused >2 SD change in "nuclear roundness", had very little impact on RelA dynamics. In my reading, this directly invalidates the core conclusion made by the authors on the ability of the actin cytoskeleton to modulate RelA dynamics.

The only drug that seemed to show a meaningful impact (my interpretation as rigorous statistical analysis of this that includes correction of multiple hypotheses is missing) was SMIFH2 which downregulates Formin activity. Naively, I googled "SMIFH2 off target" and found multiple publications suggestions that SMIFH2 has putative off target effects on other proteins such as Myosin (PMID: 33589498) but also completely unrelated targets such as P53 (PMID: 25925024). Therefore I find that the data presented in Figure 6, rather than supporting the authors' conclusion effectively invalidate it.

4. The values of RelA nuclear-cytoplasmic ratio are different across the figures in ways that I was unable to follow. In figures 1-2 the values are ~0.9-1.4. In figures 3B and 4D the same cell lines show a range of 0-3, and 0-5. Figure 3D shows RelA on a log10 scale with values are from -0.5 to 1.5 (3-33 on a linear scale). As it is presented now, this data limits the ability to draw conclusions based on RelA values across figures.

5. The authors perform cluster analysis followed by ANOVA between the clusters. This is problematic and is known as "double-dipping" and it could invalidate the statistical inference (see https://arxiv.org/abs/2012.02936). As the authors don't really use the clusters for any of their conclusions, the grouping can be avoided and analysis done directly on RelA values using correlation analysis with key features. As the authors effectively show in Figure 5A that a simple measure (RelA at 60min) is basically equivalent to cluster identity, this could substantially simplify the work and many of the panels in Figure 2 could be removed.

6. The statement made by the authors about the differences in negative feedback signaling between the two cell lines (line 153) is presented as a conclusion when in fact it is a reasonable hypothesis that is not supported by any of the data shown in the manuscript.

7. The features selected after feature selection are referred to as "independent features", yet the PCA data shown by the authors clearly show that they are not independent of each other.

8. The statement regarding the tSNE analysis (lines 279-282 and Figure 5B) is incorrect as tSNE does not preserve distance metrics and therefore proximity of points in tSNE space is not indicative of actual similarity (see: https://distill.pub/2016/misread-tsne/).

9. A more subtle version of the "double-dipping" problem mentioned in point 5 above exists in Figure 5CD. When the features tested are correlated with the features used to create the clusters, one cannot simple ANOVA between the clusters for statistical inference.

10. The inferred strengths of the arcs in the BN analysis don't fully support the authors' interpretations. For example, in Figure 4C the connection between all morphological features and RelA is very low (2-50) compared to other arcs in the same network (arc lengths are around a few hundreds with max at 4095).

11. The statistical significance of the pharmacological perturbation on RelA should be presented including controls for multiple hypothesis testing.

Specific recommendations:

The dynamic of NFkB can be assessed directly by the nuclear fraction without the cytoplasmic values. This could avoid many of the challenges related to issue 1. Care should still be taken to verify that nuclear signal doesn't vary and creates dependency, but this is much easier compared to the "ring". This is especially true for knock-ins used by the authors where there is no need to "normalized out" ectopic high overexpression levels.

Cluster analysis followed by statistical inference should be avoided and direct inference on the features used for analysis should be done instead.

*Reviewer #3 (Recommendations for the authors):*

The authors present a deep investigation of the relationship between cell morphology, actin cytoskeletal features and NF-kappaB RELA signaling dynamics. Quantitative live-cell imaging of endogenous RELA (using CRISPR knock-in) and data analyses are leveraged to provide insight into the little-understood roles of actin networks in inflammatory responses of PDAC cells to TNF-α. While the focus is clearly on cell shapes and cytoskeletal features, it will help put the study in the context of others if the PDAC cells are better characterized in terms of the negative feedback loops of NF-kappaB, such as IkappaB α or IkappaB epsilon proteins. As the authors indicated, these results altogether may provide new ideas for therapeutic interventions for PDAC.

1. As the authors mentioned in Discussion (lines 370-373 "…, other studies reported more rapid cytoplasmic REL relocalisation…"), the rapid fall of nuclear RelA (around 40 minutes after TNF-α) has widely been reported by several groups (Alexander Hoffmann, Michael White, Markus Covert, Myong-Hee Sung, Savas Tay and their colleagues), but is absent in both PDAC lines. This indicates that the negative feedback genes such as IkappaB α/epsilon or A20 may have very different kinetics if they are induced at all in these cells. It seems important to determine if this is a property of the parental PDAC cells or an acquired feature in the CRISPR knock-in reporter cells during the generation of the two reporter lines. RTqPCR of a few key negative feedback genes in the parental and derived reporter cells, as well as a control cell line (HeLa, MCF10, or THP-1) would be a straightforward way to answer this question. While this is not the focus of the study, I think that understanding which feedback is dysfunctional is probably relevant for therapeutic strategies against PDAC. It would also help the NF-kappaB signaling community understand the common and distinct features of dynamics observed in different cell systems.

2. On page 7 lines 154-155 "… negative feedback regulation is intact in PANC1…" is too simplistic, given the very slow fall of nuclear RELA even in PANC1. See comment #1 above. It looks like IkappaB feedback genes are strongly compromised in both PDAC lines. Again, this could be an important feature of PDAC and needs to be verified as suggested above. The evaluation of gene induction by RTqPCR would also be reassuring that the C-terminal fusion of RELA is transcriptionally as active as the native RELA protein, even though the EGFP is tagging the transactivation domain (important for NF-kappaB recruitment of co-factors and transcriptional machinery to the target chromatin).

3. Figure 5 has data that will be of broad interest to the community, and I was hoping to glean some recurring theme about what features may affect NF-kappaB signaling. The key results seem to be in 5C-E, but it is not easy to read off which relationships may be shared between the two PDAC lines. It will be helpful if all the ten features are shown for each cell line to see an overall pattern, even though some may fall below statistical significance. 'Actin filament/cell area' shows opposite trends in MIA PaCa2 and PANC1. It seems like 'neighbor contact' was the only feature shared by the two cell lines based on the Bayesian analysis. This was somewhat dissatisfying because it complicates any extrapolating speculations of these findings to other cell systems. But the authors seem to have done a thorough analysis using both ANOVA and Bayesian methods.

4. Cytoskeletal structures may influence rates of oligomerization or recycling of the cell surface receptors for TNF-α. Any thoughts on such indirect effects through the upstream signaling events of NF-kappaB activation?

5. The color scheme used in heatmaps (e.g. Figure 2 panels C and I) is problematic because the white is for both the strongest RelA ratio and NAs (missing values). Please use a different color scheme.

6. Figure 2-supplement 1E seems to show M1-M4 labels mixed up. Please check.

7. In line 263 "… correlation between RELA ratio and breast epithelial…", some word seems missing. Does it mean correlation between RELA ratio and "neighbor contact"?

8. In line 310, please describe what inhibitor SMIFH2 is at the first mention, for the general readers.

9. In line 313, what does "…for 2 hr then simultaneously with 10 ng/ml TNFalpha for 1 hr." mean? Consecutive treatment or simultaneous co-treament?

10. In line 420, is "NIH-T3" NIH-3T3?

[Editors’ note: further revisions were suggested prior to acceptance, as described below.]

Thank you for resubmitting your work entitled "Actin networks modulate heterogenous NF-κB dynamics in response to TNFα" for further consideration by *eLife*. Your revised article has been evaluated by Jonathan Cooper (Senior Editor) and a Reviewing Editor.

The manuscript has been improved but there are some remaining issues that need to be addressed, as outlined below:

Essential revisions (for the authors):

1) Details on the methods used for transcriptional profiling and gene expression analysis should be provided. Reviewers' requests for additional technical clarifications throughout the paper should be carefully considered and addressed where possible.

2) The method used to quantify "time to first peak" for RELA translocation should be made consistent with similar studies in the field, as noted by Reviewer 2, or should be clarified with explanations of the apparent differences relative to previous literature.

3) Inconsistencies between the text and the data shown for expression differences, as noted by Reviewer #2 (point 2) should be rectified or further explained.

4) Knockdown efficiency for the siRNA experiments should be evaluated with additional data to demonstrate the degree of heterogeneity in knockdown between cells for genes other than RELA. Alternatively, data should be provided to support the assumption that the efficiency of RELA knockdown, at the single-cell level, is representative of the other knocked-down genes.

5) Clarification of replicates and statistical significance should be provided for experiments where noted by reviewer 3.

*Reviewer #1 (Recommendations for the authors):*

In this revised manuscript, the authors have added substantial new analysis to address the primary critiques of the original manuscript. The authors have made significant changes to their image processing methods and Bayesian network modeling presentation, which in my opinion are adequate to answer the critiques raised on these points. Perhaps the largest change to the manuscript is the inclusion of an RNA-seq experiment and follow-up with knockdowns of genes of interest found in this dataset. The strongest point of this addition is that it is used to investigate the mediators of negative feedback and their differences between the two cell lines analyzed. The RNA-seq data are also used to make a connection between NF-κB activity and actin regulation, which as discussed below, makes less of an impact on the overall conclusions of the paper.

Overall, the changes do significantly strengthen the manuscript, but some revisions are still needed to fully integrate the new data.

1. By displaying NF-κB localization strictly as nuclear intensity, they avoid the difficulty of using a cytoplasmic ring to calculate nuclear/cytoplasmic intensity, a measurement that can be affected by the cell shape changes induced in their study. However, their description of the method lacks a few details – in particular, is the nuclear intensity calculated as the total RELA signal per nucleus, or the average pixel intensity over the nuclear region (line 567)? Also, the ring method is still used in the fixed cell measurements; it would be helpful to comment in more detail on the previous work that supports the insensitivity of this measurement to cell shape changes (the authors refer to Sero 2015 in their response, but it isn't clear to me where in that manuscript the ring method's response to cell shape is evaluated).

2. The addition of RNA-seq analysis provides a significant amount of new data, and the inclusion of the IkB super repressor is a nice feature of this dataset that helps increase its interpretability. However, overall this addition feels incomplete. There doesn't appear to be any description of the relevant experimental procedures in the methods section. Also, the conclusions from these experiments seem somewhat indistinct. Two actin-regulating genes, NUAK1 and ARHGEF31, are identified as targets of NF-κB, and it is shown that their knockdown modestly changes RELA translocation kinetics. However, this finding doesn't address the question of whether the actin-mediated modulation of NF-κB, which is the focus of the first part of the paper, has a functional role in altering gene expression. Thus, the last section is potentially useful, but a bit confusing in juxtaposition to the first part of the paper.

*Reviewer #2 (Recommendations for the authors):*

The authors have improved the manuscript extensively both in terms of re-analysis and re-organization of the original data and newly added experimental data and results. I appreciate the effort invested in carrying out such a major revision in response to the original comments from me and other reviewers. Many of my comments have been addressed fully or are no longer relevant in the revised version. The authors performed RNA-seq and GFP-trap experiments partly to address my concern about understanding the core regulatory circuit of NF-kappaB in these PDAC cells. However, some details are missing, and clarifications are needed in several places. While there are numerous points to commend about in this much improved version, in the interest of space, I focus here on remaining issues that need to be addressed before the manuscript can be accepted for publication.

New comments:

1. The authors note that "the time to peak RELA was highly heterogenous" (line 103). It seems like their definition is different from what other groups have been using. Instead of detecting the *time to the first peak*, it seems like the reported time to peak is catching the *time to maximum value* of nuclear RELA. This is apparent in Figure 1C (easier to see on the right side, e.g. dose response plot of MIA PaCa), where the first peak occurs earlier for the higher TNF dose (which many groups have reported to be valid in multiple cell types. For example, shown by Markus Covert, MH Sung, Michael White, etc). But the panel 1E shows the time to peak is more than 100 minutes (unit is missing in the y axis, by the way), while panel 1C shows a rapid first peak (e.g. < 30 min for 10 ng/ml TNF). I suggest the authors update their definition of time to peak and provide a more accurate quantification of the dose-dependent timing of first peak (well established in the field). With the revised definition, the time to peak may be actually not so heterogenous between individual cells (based on the data spread in panel 1C time course plots). Moreover, the time to peak of 100 minutes is not compatible with the choice of 1 hour TNF treatment in the subsequent immunofluorescence analysis. This needs to be corrected.

2. There are a few puzzling disconnects between Figure 3 Supplement panels 1A and 1A'-A'. For example, the authors interpret the data in these statements: "*NFKB2* is only affected by IκB-SR induction in PANC1 cells (Figure 3 – Supplement 1A).", "NFKB1 is unaffected by RELA-inactivation by IκB-SR in both cell lines", "NFKBIA and NFKBIB were not significantly impacted by RELA inactivation in either cell line, while NFKBIE expression was reduced with IκB-SR in PANC1". But looking at panel 1A, *NFKB2* induction is absent in MiaPaCa with IkB-SR; NFKB1 induction is also affected at 5h in both cell lines; NFKBIA induction is off but instead, the constitutive expression is higher (which indicates a high basal transcriptional activity either by NFkB or other TFs like glucocorticoid receptor). Are these simply an outcome of the statistical testing criteria? The discrepancy seems too numerous and pronounced to warrant some double-checking and/or explanation.

3. How was knockdown efficiency assessed (other than RELA) in the data of Figure 4? Individual cells may show significant heterogeneity in siRNA knockdown (which is different from the RELA siRNA), so I wonder if some of the single cell traces may be from those that didn't have a reduced abundance of the target protein (as well as the mRNA).

4. In line 122 "…no differences between cells by cell cycle stage in terms of peak RELA measurements…", it will be good to note that this is in contrast to Michael White's report on cell cycle dependence (Ankers et al. https://elifesciences.org/articles/10473). It is important to note that the crosstalk between cell cycle and NFkB is cell type specific.

5. Regarding the result "Interestingly, the NF-κB protein REL also had reduced interaction with RELA with TNFα.", there is a recent publication (Rahman SMT et al. https://www.cell.com/cell-reports/fulltext/S2211-1247(22)01556-X) reporting that the RelA:c-Rel heterodimer was depleted in the nucleus of TNF-α activated fibroblasts. This reviewer, being the senior author of the study, can't help but find this quite remarkable, and I think that the corroboration might be noteworthy in interpreting the GFP-Trap data.

6. Figure 2: The Bayesian-inferred arrows are sometimes in opposite directions between panel C and panel F (drug treatment data). Any explanations that can help readers understand would be good.

7. Figure 1-Supplement 1 panel A: Please specify that these images are from PCNA.

8. In Figure 4B legend, something seems to be missing in "(p < value)". In 4C, PCNA Scarlet is supposed to be shown.

9. Figure 4-Supplement 1: Indicating "siRNA" and "F-actin stain" in this figure would be helpful, even though they are described in the figure legend.

10. Figure 2C: In the legend, there seems to be a mix-up of "purple" for "orange" arcs in line 864.

Previous comments that still need attention (in original numbering, followed by additional comments appended after "and"):

1. As the authors mentioned in Discussion (lines 370-373 "…, other studies reported more rapid cytoplasmic REL relocalisation…"), the rapid fall of nuclear RelA (around 40 minutes after TNF-α) has widely been reported by several groups (Alexander Hoffmann, Michael White, Markus Covert, Myong-Hee Sung, Savas Tay and their colleagues), but is absent in both PDAC lines. This indicates that the negative feedback genes such as IkappaB α/epsilon or A20 may have very different kinetics if they are induced at all in these cells. It seems important to determine if this is a property of the parental PDAC cells or an acquired feature in the CRISPR knock-in reporter cells during the generation of the two reporter lines. RTqPCR of a few key negative feedback genes in the parental and derived reporter cells, as well as a control cell line (HeLa, MCF10, or THP-1) would be a straightforward way to answer this question. While this is not the focus of the study, I think that understanding which feedback is dysfunctional is probably relevant for therapeutic strategies against PDAC. It would also help the NF-kappaB signaling community understand the common and distinct features of dynamics observed in different cell systems.

and

In response to this comment, the authors "carried out RNAseq with MIA PaCa2 and PANC1 cells with endogenously tagged RELA-eGFP". However, no methods are described in the manuscript itself regarding RNA-seq, either in the Methods or figure legends. Therefore, the readers would be left guessing if the samples were from the parental PDAC or the imaged cell lines. This might be an oversight; regardless, authors need to provide the method section on RNA-seq, including sample source, replicates, sequencing platform, data processing and analysis. Also, an accession ID for the RNA-seq data should be provided after depositing the dataset to a public data repository such as GEO, which is a standard requirement, I believe.

If the RNA-seq was done on the knockin reporter cells (as stated in the response), then the analysis does not directly address the question I raised in the original comment ("…determine if this is a property of the parental PDAC cells or an acquired feature in the CRISPR knock-in reporter cells during the generation of the two reporter lines"). Barring another round of RNA-seq analysis on the parental cell lines, for a minimum effort, the authors can perform RTqPCR of NFKBIA, NFKBIE, REL, RELB, *NFKB2* (those found to be regulated by RelA in the reporter cells) in both the parental and the reporter cells. This would confirm that the observed NF-κB pathway gene expression patterns are indeed a property of PDAC cells.

4. Cytoskeletal structures may influence rates of oligomerization or recycling of the cell surface receptors for TNF-α. Any thoughts on such indirect effects through the upstream signaling events of NF-kappaB activation?

and

The author response to this comment contains quite extensive information. The content, or a brief summary, seems to warrant inclusion in the main text of the paper, either in the Results or Discussion. This is an aspect that is not the focus of the study but may be acknowledged as complementary mechanisms to be explored in future studies. Such a bigger picture discussion might encourage readers to explore open topics in their own studies.

*Reviewer #3 (Recommendations for the authors):*

A revised manuscript presents the study of NF-κB signalling in PDAC tumours. The authors demonstrate that the TNF-induced responses of the canonical p65 signalling are mediated via F- actin dynamics. I believe this represents a novel and important finding.

New data in the revised manuscript provide an analysis of TNF-induced gene expression via RNA-seq and identify specific feedback mechanisms, involving IkappaB inhibitors and family members (RelB) as well as actin regulators NUAK2 and ARHGAP31. Furthermore, the authors perform siRNA knockdown experiments to validate specific targets, which provide an excellent contribution to the narrative.

The manuscript is presented well and analyses are performed to a high standard. However, the manuscript suffers from some ambiguities regarding sample sizes and statistical analyses. Firstly, most if not all data is not triplicated, instead duplicates while some technical replicates are presented (e.g., Figure 1, Figure 3, Figure 3 S1, Figure 4) -sometimes no information at all is provided (Figure 4C, Figure 4 S1). While part of this info is provided in methods, these should be included in appropriate captions. Secondly, there are limited details in terms of the statistical analyses, typically t-tests are performed (with corrections for multiple testing), often in the case of seemingly small sample sizes. E.g. Figure 3 S1 A' A'- t-test based on 2 samples (as far I understand). T critical he siRNa KO experiment (Figure 4) seems to involve 6 or 8 samples (judging by eye Figure A and B), for which typically a non-parametric test should be used instead. Given that the effect of siRNa-KO (in particular of the actin regulator genes) on p65 dynamics in the live-cell imaging data is subtle (Figure 4A), this poses questions about whether the conclusions are robust.

Specific comments:

1. Line 41: "However, most studies characterising RELA translocation dynamics following stimulation use hyperphysiological TNFα doses (e.g. 10 ng/ml) and exogenous RELA reporters." This sentence is not accurate…both have been studied for more than 10 years, e.g. see [1-3],

2. Lines 106-112: Authors report correlations between different times, are they suggesting that total p65 is regulated? Please explain or provide the measurement of the total p65 over time.

3. RELA translocation responses to TNFα are cell cycle independent: Please discuss in the context of the previous work on this subject [4].

4. Figure 1 and S1 have no description of statistical tests

5. Figure 2C: numbers characterising the strength of the relationship span over 3 -orders of magnitude. Please comment on their statistical significance…

6. 2F. Please provide some validation that chemical perturbation causes measurable changes in any of the cell features…in addition to affecting high-level Bayesian analysis. 2F please provide how many cells analysed

7. Figure 3 A. How many replicates were assayed, and how the information about different time points is provided on the graph?

8. Figure 3 S1 A' A'- Statistical analysis using t-test based on 2 samples (as far I understand). (also line 283 in the text).

9. Figure 4 "To identify whether siRNAs affected the early/peak RELA and sustained RELA response to TNFα, we calculated the fold change of mean nuclear RELA with each siRNA to NT siRNA at 1 hr or 12 hr TNFα stimulation, and compared fold changes using t-test with multiple comparison correction (Figure 4B and 4C)."

The effect of NUAK2 and ARGHAP31 siRNA KO appears to be subtle in live cell imaging from A- is there a statical difference in the AUC or any other characteristics of p65 responses?

Analysis in 4B is critical, but not clear to me what is the sample size (how many wells…) and how matching was performed. By eye, it seems that the sample size is between 6-8, but please explicitly provide the number in the legend. T-test with small sample size is not appropriate, a non-parametric test should be used instead.

10. Figure 4C and Supplementary info- no information about data replication -just some images are shown.

11. Line 354: MIA PaCa2 and PANC1 cells in the presence of ARHGAP31 siRNA showed flatter morphology and reduction of stress fibre abundance, while NUAK2 siRNA visibly increased actin abundance and the presence of lamellipodia in both cell lines. -> These claims should be statistically tested based on the replicated data.

12. Line 318: (Figure 4A-D) -there is no D in the figure

Refs:

1. Sung MH, Salvatore L, De Lorenzi R, Indrawan A, Pasparakis M, Hager GL, Bianchi ME, Agresti A: Sustained oscillations of NF-kappaB produce distinct genome scanning and gene expression profiles. PLoS One 2009, 4:e7163.

2. Tay S, Hughey JJ, Lee TK, Lipniacki T, Quake SR, Covert MW: Single-cell NF-kappaB dynamics reveal digital activation and analogue information processing. Nature 2010, 466:267-271.

3. Turner DA, Paszek P, Woodcock DJ, Nelson DE, Horton CA, Wang Y, Spiller DG, Rand DA, White MR, Harper CV: Physiological levels of TNFalpha stimulation induce stochastic dynamics of NF-kappaB responses in single living cells. J Cell Sci 2010, 123:2834-2843.

4. Ankers JM, Awais R, Jones NA, Boyd J, Ryan S, Adamson AD, Harper CV, Bridge L, Spiller DG, Jackson DA, et al: Dynamic NF-kappaB and E2F interactions control the priority and timing of inflammatory signalling and cell proliferation. *ELife* 2016, 5.

---

## [Author Response]

[Editors’ note: the authors resubmitted a revised version of the paper for consideration. What follows is the authors’ response to the first round of review.]

Essential revisions:Reviewer #1 (Recommendations for the authors):In this study, the authors set out to profile the kinetics of RELA translocation in pancreatic cancer cells. RELA translocation is a widely studied event that occurs in response to a number of inflammatory stimuli, and the authors begin by profiling RELA translocation kinetics in two PDAC cell lines. Relative to other cell lines where this analysis has been performed, the two PDAC cell lines show more sustained signaling (to different extents), rather than oscillatory nuclear localization. Like many other cell lines that have been studied, the authors observe a large degree of heterogeneity in RELA translocation within cells stimulated with physiological doses of TNF. A thorough analysis based on a knock-in PCNA marker reveals that this heterogeneity is not linked to the cell cycle. The authors then hypothesize the connection between actin morphology based on their previous work, and turn to an analysis of 5 PDAC cell lines, using a fixed-cell immunofluorescence assay to quantify RELA translocation, F-actin, and tubulin in each cell. From images of thousands of fixed cells, a series of statistical analyses are used to identify correlations between cytoskeletal morphology and RELA nuclear abundance. This approach reveals clear associations between certain cytoskeletal features and RELA translocation, with substantial differences between cell lines. A follow-up experiment correlates live-cell RELA kinetics with fixed-cell cytoskeletal stains, which provides further support for their associations. Finally, they perturb actin and tubulin function with a variety of chemical agents and quantify the resulting effects on live-cell RELA kinetics. This again shows variation between cell lines but confirms that direct perturbation of cytoskeletal processes can modulate the ability of TNF to stimulate RELA translocation. Together, these lines of evidence provide substantial support for the hypothesis that NF-κB signaling can be modulated downstream of changes in actin dynamics. While there is significant variation in how this works between cell lines, which suggests that a broader exploration of other cell lines will be needed in the future, the data shown here are, with some limitations noted below, adequate to support the authors' hypothesis.

We thank reviewer 1 for the detailed reading of our manuscript and for their positive and insightful evaluation of our work.

A major strength of this manuscript is its use of quantitative imaging to exploit the existing heterogeneity between cells at a physiological dose of TNF. A series of clever analysis methods are used to draw out correlations that can be found between actin morphology (immunostained) and RELA translocation (either immunostained or by live-cell microscopy). Also important is the careful characterization of NF-κB signaling in pancreatic cancer cells, where this pathway is known to play a role. Importantly, RELA kinetics are assessed using a knock-in fluorescent protein tag, avoiding potential artifacts from overexpression. An interesting feature of this analysis is the use of Bayesian networks to model the dependencies of actin features; to my knowledge, this is a novel use of this algorithm. The drug perturbation data are also beautifully presented, in a way that reveals the full complexity of the cellular responses. In general, the data and conclusions are very clearly presented, and the authors do not shy away from describing the context-dependent intricacies of their observations. The main conclusion that NF-κB signaling is strongly affected by actin dynamics ties together two major signaling axes in cancer cells, and would be of interest to the many researchers working in these two areas. The methods used to show associations at the single-cell level will also have a significant impact on others working with mixtures of live and fixed single-cell data.

Thank you to Reviewer 1 for their accurate summary of our manuscript and comments on the manuscript’s wider significance. We agree that our manuscript will be of particular interest to the large communities of researchers working on the cytoskeleton or NF-κB, as well as the single cell dynamics audience researching alternative pathways.

There are several weaknesses that limit the interpretability of the findings:1. It is not clear how the effects of cell shape changes on the recorded nuclear/cytoplasmic ratio are controlled for. Cell shape is a major variable in this study and can alter the apparent (recorded) intensity of nuclear and cytoplasmic stains in many imaging and analysis pipelines, independent of any actual changes in a marker's localization. Without such an assessment, it isn't clear whether differences in RELA behavior (the main observation in the paper) are influenced by how the detection process treats cells with different morphologies.

We appreciate the reviewer’s concern that signal intensity in the cytoplasm could be affected by cell morphology, e.g. thicker cells could generate higher cytoplasmic intensity. We justified use of the ring region to account for potential issues with cytoplasmic measurements, since we previously showed that the RELA signal in the ring region of the cytoplasm is not affected by cell morphology (Sero et al., 2015).

Nonetheless, as we have generated endogenously tagged RELA-GFP in the present study with minimal cell-to-cell variability in RELA-GFP expression in basal conditions, we have altered our manuscript to present the raw nuclear RELA-GFP intensity measurements without normalisation to ring or cytoplasmic measurements. Importantly, removal of normalisation to the ring region has not affected any of the major conclusions in this manuscript and raw RELA measurements with TNFα still portray sustained and non-oscillatory single cell dynamics (revised Figure 1C).

2. It is not clear whether the cytoskeleton-linked effects are specific to RELA, or rather reflect more widespread changes in nuclear import/export rates. There are reports that changes in actin function can alter nuclear-cytoplasmic translocation (for example, PMIDs 16120220, 31444357). Understanding whether the observed actin-dependent effect on NF-κB localization is unique to this signaling pathway, or a non-specific effect on multiple pathways is important to the interpretation of the findings.

We are also interested in whether actin regulation of RELA is specific to RELA or applicable to transcription factors in general. We have previously shown that nuclear to ring region measurements of the transcription co-activator YAP correlate with cell shape features that did not also correlate with the transcription factor JUN (Sero et al., 2015), however, these analyses did not include actin measurements. This could be tested by co-staining for other transcription factors, e.g. p53 or STATs, and assessing dependencies of the transcription factor localisation on actin features. However, these experiments are not necessary to draw conclusions about RELA regulation by actin, which is one of the major focuses of this manuscript.

3. It is difficult to assess the importance of the observed differences in RELA translocation without additional information on the pathway's output. Translocation of RELA is an important marker and is known to modulate many downstream genes involved in inflammation. However, given the differences in RELA behavior observed here in PDAC cells, it is difficult to extrapolate from previous work to evaluate whether the actin-modulated changes impact RELA's transcriptional effects. Establishing that such differences exist would be important to conclude that actin shape changes have a biologically significant effect on NF-κB signaling with the potential to alter cellular behavior.

We approached whether cell shape or cytoskeletal features regulate RELA signalling using image-based analysis of nuclear RELA intensity as RELA subcellular localisation is a major determinant of its transcriptional output. We agree with Reviewer 1 that assay of RELA transcriptional output in PDAC cells is both interesting and necessary to understand the output of RELA in PDAC. We have carried out RNA-seq with PDAC cells under various TNFα concentrations, in the presence or absence of non-degradable IκB, to identify potential RELA transcriptional targets in PDAC. These data are included in our revised manuscript (below).

‘As RELA is known to be involved in feedback loops with transcriptional targets, with the best studied example being IκBα, we sought to identify what the transcriptional targets of RELA are in PDAC and which influence RELA in actin dependent and independent ways.

To this end, we carried out RNA-seq analysis of PDAC cells at an early (1 hr) and late (5 hr) timepoint with varying TNFα doses (0.01, 0.1 or 10 ng/ml TNFα) using MIA PaCa2 and PANC1 cells. TNFα treatments were in the presence or absence of dox-induction of IκB super repressor (IκB-SR) to determine which genes require RELA nuclear translocation. 254 genes were differentially expressed in high TNFα (0.1 or 10 ng/ml) versus basal conditions, with the majority of genes (186) upregulated by TNFα (Figure 3A and 3B). To identify patterns of gene expression across TNFα doses and durations, the 254 TNFα-regulated genes were organised by hierarchical clustering into seven clusters, each with distinct expression dynamics or dependence on RELA (Figure 3C and 3D).’

Although we have not assayed differences in gene expression with actin interference, our revised manuscript does include identification of two RELA targets (NUAK2 and ARHGAP31) that are involved in actin modulation and also perturb RELA dynamics (Author response image 1), therefore identifying components of regulatory network actin and RELA in PDAC. However, we agree that gene expression analysis would certainly add to our findings by live imaging.

‘We then focused on the effect of siRNA knockdown of the putative actin regulators and TNFα/RELA-dependent genes ARHGAP31 and NUAK2 on early (1 hr post-TNFα) and late (12 hr post-TNFα) nuclear RELA in MIA PaCa2 and PANC1 cells, in order to explore the dependence of RELA on actin dynamics computationally predicted by Bayesian modelling. Strikingly, ARHGAP31 depletion significantly increased the early (1 hr) response of PANC1 cells to TNFα and the late (5 hr) response of MIA PaCa2 to TNFα, while NUAK2 depletion increased nuclear RELA with TNFα in MIA PaCa2 cells (Figure 4B and 4C). Thus, NUAK2/ARHGAP31 expression impacts TNFα-mediated RELA dynamics in PDAC cells.’

**Author response image 1. sa2fig1:** 

Reviewer #2 (Recommendations for the authors):The work by Butera et al. investigates the relationship between the cellular cytoskeleton and NFkB dynamic response to TNF. The authors use single-cell fixed and live cells measurements of RelA dynamics and cellular morphology, pharmacological perturbations, and extensive statistical analysis to draw specific conclusions on what aspects of the actin cytoskeleton modulate TNF-dependent NFkB response. I have identified multiple issues that raise concerns that the conclusions of this paper are not supported by the data presented.1. The possibility of confounding factors between cell morphology and the approach used by the authors to analyze NFkB dynamics (Figure 1D). The analysis of NFkB dynamics presented here is based on an image segmentation strategy that relies on a single marker, PCNA, and segmentation of the nucleus and "ring" around the nucleus. This strategy should only be applied in contexts where cell shape is not changing and is not a key factor in the investigation and it cannot be used to investigate the dependency of NFkB on cell shape. The main issue is that the "ring" is not a good proxy for the actual concentration of the fluorescent marker in the cytoplasm. For example, in a case where the marker (RelA-GFP in this case) is not changing, the measurement of the "cytoplasmic" signal can change as the cell spreads from a rectangular shape with limited lamellipodia to a cell that is very spread with thin lamellipodia. This change does not mean that the concentration of the marker in the cytoplasm changed, it just reveals the use of "ring" as a proxy is limited. This puts the key measurement of this work in doubt as it is the key theme throughout this paper. To give just one example, as the number of "neighbor contact" increases, cells tend to have thicker and less spread morphology (i.e. they are just denser in culture). Thicker cells will have more cytoplasm in the "ring" compared to very spread cells and therefore the denominator of RelA will be bigger that will cause the overall RelA ratio to be smaller which is exactly what the authors show in Figure 4D. Just to be clear, this is not just an issue with cell thickness, the estimation of total cytoplasmic intensity from a "ring" inherently depends on cell morphology. Therefore, one simply cannot state that RelA activation itself depends on cell morphology using this analysis method.

We thank reviewer 2 for their critical assessment of the segmentation and image analysis strategies used in this manuscript and their fit for morphological data. Given the use of endogenously tagged RELA in this study, we agree that normalisation to the ring region is not needed, and have changed all figures to present nuclear RELA measurements.

2. Bayesian network analysis does not provide a causal relationship. The authors make extensive use of Bayesian networks (BN) for inferences of dependencies (Figures 3 and 4). In this analysis, the authors aim to identify the direction of dependency between different nodes in the BN. This is problematic for two reasons. First, The direction of arrows in bayesian networks does not imply causations (i.e. modulation). The direction of arrows in BN only represents correlation and the directionality should not be interpreted as causative. Yet the authors clearly make a key distinction whether RelA depends on a morphological feature or whether the feature depends on RelA. In the literature that the authors cite (e.g. Sachs et al. 2005) pharmacological perturbation is used to infer causation, not the inference of the structure of the BN. Second, the authors did not consider that other factors, not included in the model, could influence both RelA and morphology. For example, if an unknown kinase changes both NFKB and a specific cellular feature the inferred network will still look the same. Therefore, the conclusion that actin-based cellular features modulate (i.e. cause) differences in RelA dynamics is simply unsupported by the BN analysis.

We thank reviewer 2 for outlining the roles and limitations of Bayesian network analysis and for pointing out the necessity for clear language when describing results from Bayesian networks. We acknowledge that arcs presented in Bayesian networks are indicative of probabilistic causality, and we did not intend to imply that these are direct mechanistic biological interactions, but sought to present statistically likely dependencies that generate hypotheses. Based on reviewer 2’s comments, we have challenged the interactions between RELA and actin, tubulin and cell shape features identified our original Bayesian models by constructing Bayesian networks using systematic perturbation of PDAC cells with diverse small molecule inhibitors targeting the cytoskeleton (revised Figure 3B):

‘When using datasets of sufficient size, deriving Bayesian models based on largely stochastic and relatively small fluctuations in variables can provide insight into the influence of cytoskeletal components, shape, and RELA on each other (Figure 2C). But molecular interventions provide a means to further drive ordering of connections by identifying regions in state where normally correlated variables become conditionally independent (Pe’er et al., 2001, Sachs et al., 2005). Consequently, we created an additional dataset where cells were imaged following perturbations of tubulin, actin, myosin, or focal adhesion (FA) dynamics (Figure 2E). Such perturbations are intended to alter the value of variables/features (i.e. measures of actin organization) beyond those observed in normal populations.’

‘We generated a single cell dataset for use in Bayesian modelling by treating MIA PaCa2 and PANC1 cells with selected drug doses for 2 hr then simultaneously with TNFα (0, 0.01, 0.1 and 10 ng/ml) for 1 hr and input these data into the same Bayesian algorithm as above (rsmax2) (Figure 2F). Interestingly, the Bayesian network following perturbations revealed that 5/6 of the variables observed to correlate with RELA (whether they influenced RELA or were influenced by RELA) in untreated cells were independent of RELA in the drug-treated network. Only ‘Cytoplasmic Actin Mean’ remained as a influencing variable following drug-treatment, suggesting that actin abundance is a key regulator of RELA nuclear localisation. While other variables (i.e. tubulin and cell shape) can indirectly influence RELA, they do so via regulating actin network organisation.’

We have additionally revised the text in our manuscript to make our original conclusions more clear and emphasised the need for heterogeneity in order to make computational predictions:

‘In order to provide the greatest heterogeneity in nuclear RELA and cytoskeletal/cell shape features for Bayesian model generation, we collated data from all five PDAC lines and TNFα doses (0, 0.01, 0.1 and 10 ng/ml), shown in Figure 2C. This model indicated that nuclear RELA measurements are correlated to and predicted to be dependent on cytoplasmic actin and tubulin intensity, suggesting that cytoskeletal dynamics influence heterogeneity in nuclear RELA translocation with TNFα. Nuclear RELA is also predicted to be dependent on cell area, although with a lower strength of the probabilistic relationship compared to actin/tubulin, while nucleus roundness is predicted to be dependent on nuclear RELA. Interestingly, both actin texture and the ratio of membrane to cytoplasm actin are also predicted to be dependent on RELA, suggesting that RELA and actin dynamics are interdependent.’

3. Lack of support by pharmacological data. Beyond correlation analysis, the authors used a pharmacological approach to show that manipulation of the cytoskeleton causes changes in RelA dynamics. The data presented in Figures 1-5 makes specific predictions. If one is to interpret the arrows in the BN as the authors present them, one will conclude that perturbations of features like "nuclear roundness" that was presented as key "modulator" of RelA dynamics will cause a change in RelA dynamics. The vast majority of the pharmacological perturbations used by the authors, even those that caused >2 SD change in "nuclear roundness", had very little impact on RelA dynamics. In my reading, this directly invalidates the core conclusion made by the authors on the ability of the actin cytoskeleton to modulate RelA dynamics.

Reviewer 2 raises several excellent points regarding the limitations of the interpretations of Bayesian networks. We agree that the small molecules used to perturb the cytoskeleton have varied and limited impact on RELA dynamics, and recognise that the small molecules used target several pathways/cell shape components, which may contribute to the confounded effect on RELA. Therefore, we have removed any focus of individual small molecules on nuclear RELA and have collated the small molecule perturbation data into a Bayesian network model, in line with Reviewer 2’s other comments. In addition to inhibitor treatments, we have carried out an orthogonal experiment to test the effect of depletion by siRNA of specific genes (ARHGAP31 and NUAK2) involved in actin modulation to test potential regulation of RELA by actin dynamics, as hinted computationally by our computational analysis. We believe that this experiment provides more conclusive evidence for a regulatory network between actin and RELA.

The only drug that seemed to show a meaningful impact (my interpretation as rigorous statistical analysis of this that includes correction of multiple hypotheses is missing) was SMIFH2 which downregulates Formin activity. Naively, I googled "SMIFH2 off target" and found multiple publications suggestions that SMIFH2 has putative off target effects on other proteins such as Myosin (PMID: 33589498) but also completely unrelated targets such as P53 (PMID: 25925024). Therefore I find that the data presented in Figure 6, rather than supporting the authors' conclusion effectively invalidate it.

We thank reviewer 2 for outlining potential issues with the use of the formin inhibitor SMIFH2. Following revision of our manuscript, we have omitted emphasis on formin and RELA as we have used the small molecule data to provide perturbation and heterogeneity for Bayesian modelling of RELA and cell feature interactions. Our repurposing of these date removes the importance of any single small molecule effects on nuclear RELA.

4. The values of RelA nuclear-cytoplasmic ratio are different across the figures in ways that I was unable to follow. In figures 1-2 the values are ~0.9-1.4. In figures 3B and 4D the same cell lines show a range of 0-3, and 0-5. Figure 3D shows RelA on a log10 scale with values are from -0.5 to 1.5 (3-33 on a linear scale). As it is presented now, this data limits the ability to draw conclusions based on RelA values across figures.

We have amended figures to present RELA quantification on the same scale (unlogged).

5. The authors perform cluster analysis followed by ANOVA between the clusters. This is problematic and is known as "double-dipping" and it could invalidate the statistical inference (see https://arxiv.org/abs/2012.02936). As the authors don't really use the clusters for any of their conclusions, the grouping can be avoided and analysis done directly on RelA values using correlation analysis with key features. As the authors effectively show in Figure 5A that a simple measure (RelA at 60min) is basically equivalent to cluster identity, this could substantially simplify the work and many of the panels in Figure 2 could be removed.

Reviewer 2 is correct in that clustering does not provide additional information to this dataset. We have omitted clustering from analysis of the RELA ratio tracks, which is now categorised by TNFα treatment alone (Figure 1C from the revised manuscript).

6. The statement made by the authors about the differences in negative feedback signaling between the two cell lines (line 153) is presented as a conclusion when in fact it is a reasonable hypothesis that is not supported by any of the data shown in the manuscript.

We agree with the reviewer’s comment that data on negative feedback proteins is necessary to make conclusions regarding the two cell lines. To this end, we have carried out Co-IP to assess RELA interaction with IκB proteins (the key negative regulators of RELA) and additionally performed RNA-seq to assess the expression of IκB genes in MIA PaCa2 and PANC1 cells. Furthermore, we have depleted IκB genes in MIA PaCa2 and PANC1 cells and assessed the effects on RELA dynamics. These results have been incorporated into the revised manuscript.

7. The features selected after feature selection are referred to as "independent features", yet the PCA data shown by the authors clearly show that they are not independent of each other.

We chose cell features based on lack of intercorrelation (revised Figure 2 – Supplement 1B) but appreciate the reviewer’s comment that this is insufficient to refer to the as features ‘independent’ when they have dependencies shown in the Bayesian network models. Accordingly, we have removed the term ‘independent’ when referring to the features selected.

8. The statement regarding the tSNE analysis (lines 279-282 and Figure 5B) is incorrect as tSNE does not preserve distance metrics and therefore proximity of points in tSNE space is not indicative of actual similarity (see: https://distill.pub/2016/misread-tsne/).

Thank you to Reviewer 2 for highlighting this. We have omitted the tSNE analysis.

9. A more subtle version of the "double-dipping" problem mentioned in point 5 above exists in Figure 5CD. When the features tested are correlated with the features used to create the clusters, one cannot simple ANOVA between the clusters for statistical inference.

We thank the reviewer for drawing attention to the double dipping in the original Figure 5C/D. We have removed all clustering of RELA data and associated statistical analysis.

10. The inferred strengths of the arcs in the BN analysis don't fully support the authors' interpretations. For example, in Figure 4C the connection between all morphological features and RelA is very low (2-50) compared to other arcs in the same network (arc lengths are around a few hundreds with max at 4095).

The reviewer has drawn attention to an important discussion point regarding Bayesian modelling and the significance of arc strengths. We agree that the connections involving RELA in our Bayesian models have low arc strengths when compared to arcs in the same network. However, we expected that the arc strengths between cytoskeletal and cell shape features would be exceptionally high, given how linked they are biologically. We therefore did not aim to compare the arc strengths involving RELA to the arc strengths only involving cytoskeletal and cell shape features, and aimed to use a relatively stringent algorithm to construct our Bayesian models (rsmax2). Nonetheless, we hope that our revised text reflects more considerate language when referring to the conclusions made from Bayesian networks in general, as encouraged by Reviewer 2’s comments.

11. The statistical significance of the pharmacological perturbation on RelA should be presented including controls for multiple hypothesis testing.

We thank Reviewer 2 for the important reminder to present statistical significance and highlight use of multiple test correction. As mentioned, we have omitted these figure sections altogether, however, they can be re-included (with statistical significance) as a part of the main figure or as a supplement, if the reviewers request.

Reviewer #3 (Recommendations for the authors):The authors present a deep investigation of the relationship between cell morphology, actin cytoskeletal features and NF-kappaB RELA signaling dynamics. Quantitative live-cell imaging of endogenous RELA (using CRISPR knock-in) and data analyses are leveraged to provide insight into the little-understood roles of actin networks in inflammatory responses of PDAC cells to TNF-α. While the focus is clearly on cell shapes and cytoskeletal features, it will help put the study in the context of others if the PDAC cells are better characterized in terms of the negative feedback loops of NF-kappaB, such as IkappaB α or IkappaB epsilon proteins. As the authors indicated, these results altogether may provide new ideas for therapeutic interventions for PDAC.

Thank you to Reviewer 3 for this concise summary of our manuscript. We agree that further enquiry into IκB proteins is necessary to fully characterise RELA signalling in PDAC cells and to aid comparison with other cell lines.

Inspired by the reviewers’ comments, we performed RNA-seq assessment of genes encoding IκB proteins (revised Figure 3 – Supplement 1A).

In addition, we used GFP-Trap in PDAC cells expressing endogenously tagged RELA-GFP to identify the binding partners of RELA in PDAC cells and analysed changes in protein abundance with TNFα addition by mass spectrometry:

‘Lastly, we checked which NF-κB/IκB proteins interact with RELA in PDAC cells. GFP-Trap followed by mass spectrometry with MIA PaCa2 cells expressing endogenously tagged RELA-GFP pulled down six NF-κB/IκB proteins (Figure 3 – Supplement 1B). IκBα/NFKBIA, IκBβ/NFKBIB and IκBε/NFKBIE were pulled down at lower abundance with increasing TNFα dose, in line with the well-studied degradation of IκB proteins downstream of TNFα stimulation (Chen et al., 1995). Interestingly, the NF-κB protein REL also had reduced interaction with RELA with TNFα. In contrast, NFKB1 and *NFKB2* showed increased interaction with RELA with TNFα. RELA therefore appears to form NF-κB heterodimers with REL, NFKB1 and *NFKB2* in PDAC cells.’

1. As the authors mentioned in Discussion (lines 370-373 "…, other studies reported more rapid cytoplasmic REL relocalisation…"), the rapid fall of nuclear RelA (around 40 minutes after TNF-α) has widely been reported by several groups (Alexander Hoffmann, Michael White, Markus Covert, Myong-Hee Sung, Savas Tay and their colleagues), but is absent in both PDAC lines. This indicates that the negative feedback genes such as IkappaB α/epsilon or A20 may have very different kinetics if they are induced at all in these cells. It seems important to determine if this is a property of the parental PDAC cells or an acquired feature in the CRISPR knock-in reporter cells during the generation of the two reporter lines. RTqPCR of a few key negative feedback genes in the parental and derived reporter cells, as well as a control cell line (HeLa, MCF10, or THP-1) would be a straightforward way to answer this question. While this is not the focus of the study, I think that understanding which feedback is dysfunctional is probably relevant for therapeutic strategies against PDAC. It would also help the NF-kappaB signaling community understand the common and distinct features of dynamics observed in different cell systems.

We agree that it is necessary to show functionality of RELA following C-terminal addition of eGFP and also interesting to identify which negative feedback genes are important in PDAC cells. To this end, we carried out RNAseq with MIA PaCa2 and PANC1 cells with endogenously tagged RELA-eGFP and checked upregulation of IκB-encoding genes, in addition to NF-κB family genes, which we are in revised Figure 3 – Supplement 1A. Interestingly, we find that expression of the IκBε-encoding gene *NFKBIE* is dependent on TNFα dose and on RELA nuclear translocation, since *NFKBIE* expression is abrogated in the presence of non-degradable IκBα (IκB-SR) in both PDAC cell lines.

‘RNA-seq in MIA PaCa2 and PANC1 cells revealed that expression of *RELA* itself did not significantly scale with TNFα dose, while *RELB* and the transcriptionally incompetent NF-κB proteins *NFKB1* and *NFKB2* showed increasing expression with increasing TNFα (Left graph in Figure 3 – Supplement 1A). TNFα addition significantly altered the expression of a subset of IκB protein-encoding genes, which display distinct expression dynamics: *NFKBIA* (Figure 3: Cluster 5), *NFKBIB* (Figure 3: Cluster 4), *NFKBIE* (Figure 3: Cluster 3), *NFKBID* (Figure 3: Cluster 6) and *NFKBIZ* (Figure 3: Cluster 6). Of these, *NFKBIA* has the highest absolute RNA expression in MIA PaCa2 cells and *NFKBIB* has the highest abundance in PANC1 cells, while *NFKBID* and *NFKBIZ* have the lowest RNA expression in both cell lines (right graph in Figure 3 – Supplement 1A).

T-test comparison with multiple test correction of RNA expression in TNFα-treated cells revealed that *RELB* and *REL* have significantly reduced expression with IκB-SR presence versus absence in both MIA PaCa2 and PANC1 cells, while *NFKB2* is only affected by IκB-SR induction in PANC1 cells (Figure 3 – Supplement 1A’).

Furthermore, expression of the canonical RELA binding partner *NFKB1* is unaffected by RELA-inactivation by IκB-SR in both cell lines. Similarly, expression of the IκB family protein-encoding genes *NFKBIA* and *NFKBIB* were not significantly impacted by RELA inactivation in either cell line, while *NFKBIE* expression was reduced with IκB-SR in PANC1. Therefore, of the core NF-κB signalling components, *RELB*, *REL* and *NFKBIE* expression appear to be dependent on TNFα-stimulated RELA nuclear translocation in PDAC cells. We also considered how expression of NF-κB and IκB genes are affected by the treatment duration of TNFα (Figure 3 -Supplement 1A’’), combining data from 0.1 ng/ml and 10ng/ml TNFα. In both MIA PaCa2 and PANC1 cells, *RELB* expression was significantly higher with 5 hr compared to 1 hr TNFα, which was also the case for *NFKB2* and *NFKBIE* in the PANC1 cell line alone.’

Using RNAseq, we also identified multiple known RELA targets as upregulated by TNFα only in the absence of IκB-SR in the PDAC cell lines, including *RELB*, *NFKB2*, *NFKBIE*, and *TNFAIP3* (A20), which we have included in the revised manuscript (Cluster 3 genes in Figure 3):

We believe these data in our revised manuscript verify intact functioning of RELA in the two RELA reporter PDAC cell lines and provide previously unknown data on negative feedback signalling in PDAC cells.

2. On page 7 lines 154-155 "… negative feedback regulation is intact in PANC1…" is too simplistic, given the very slow fall of nuclear RELA even in PANC1. See comment #1 above. It looks like IkappaB feedback genes are strongly compromised in both PDAC lines. Again, this could be an important feature of PDAC and needs to be verified as suggested above. The evaluation of gene induction by RTqPCR would also be reassuring that the C-terminal fusion of RELA is transcriptionally as active as the native RELA protein, even though the EGFP is tagging the transactivation domain (important for NF-kappaB recruitment of co-factors and transcriptional machinery to the target chromatin).

As above, we checked that C-terminal fusion eGFP to RELA does not impact the transcriptional functionality of RELA using RNAseq (Figure 3 in revised manuscript). We also checked using co-immunoprecipiation that RELA-eGFP binds to NF-κB and IκB proteins (Figure 3 – Supplement 1B):

‘GFP-Trap followed by mass spectrometry with MIA PaCa2 cells expressing endogenously tagged RELA-GFP pulled down six NF-κB/IκB proteins (Figure 3 – Supplement 1B). IκBα/NFKBIA, IκBβ/NFKBIB and IκBε/NFKBIE were pulled down at lower abundance with increasing TNFα dose, in line with the well-studied degradation of IκB proteins downstream of TNFα stimulation (Chen et al., 1995). Interestingly, the NF-κB protein REL also had reduced interaction with RELA with TNFα. In contrast, NFKB1 and *NFKB2* showed increased interaction with RELA with TNFα. RELA therefore appears to form NF-κB heterodimers with REL, NFKB1 and *NFKB2* in PDAC cells.’

3. Figure 5 has data that will be of broad interest to the community, and I was hoping to glean some recurring theme about what features may affect NF-kappaB signaling. The key results seem to be in 5C-E, but it is not easy to read off which relationships may be shared between the two PDAC lines. It will be helpful if all the ten features are shown for each cell line to see an overall pattern, even though some may fall below statistical significance. 'Actin filament/cell area' shows opposite trends in MIA PaCa2 and PANC1. It seems like 'neighbor contact' was the only feature shared by the two cell lines based on the Bayesian analysis. This was somewhat dissatisfying because it complicates any extrapolating speculations of these findings to other cell systems. But the authors seem to have done a thorough analysis using both ANOVA and Bayesian methods.

Thank you to reviewer 3 for their interest in the features correlated with nuclear RELA in PDAC cells. We have summarised the relationships between cytoskeletal/cell shape features and RELA, identified by Bayesian modelling, in revised Figure 2D. As requested, each facet of this figure shows a cell feature and the top row of each facet uses data from all PDAC cell lines, which informs on whether the relationship between the cell feature and RELA is shared among the five PDAC cell lines tested. Notably, Bayesian modelling identifies that heterogeneity in nuclear RELA intensity is dependent on cytoplasm actin intensity and cytoplasm tubulin intensity across PDAC cell lines, suggesting that cytoskeletal regulation of RELA is a recurring phenomenon.

‘In order to provide the greatest heterogeneity in nuclear RELA and cytoskeletal/cell shape features for Bayesian model generation, we collated data from all five PDAC lines and TNFα doses (0, 0.01, 0.1 and 10 ng/ml), shown in Figure 2C. This model indicated that nuclear RELA measurements are correlated to and predicted to be dependent on cytoplasmic actin and tubulin intensity, suggesting that cytoskeletal dynamics influence heterogeneity in nuclear RELA translocation with TNFα. Nuclear RELA is also predicted to be dependent on cell area, although with a lower strength of the probabilistic relationship compared to actin/tubulin, while nucleus roundness is predicted to be dependent on nuclear RELA. Interestingly, both actin texture and the ratio of membrane to cytoplasm actin are also predicted to be dependent on RELA, suggesting that RELA and actin dynamics are interdependent.’

4. Cytoskeletal structures may influence rates of oligomerization or recycling of the cell surface receptors for TNF-α. Any thoughts on such indirect effects through the upstream signaling events of NF-kappaB activation?

Reviewer 3 raises an interesting question that highlights the current gap in understanding of cytoskeletal regulation of TNF receptor stability and conformation. In literature, there is evidence for modulation of nuclear RELA by actin and tubulin inhibitors (Bourgarel-Rey et al., 2001, Molecular Pharmacology; Németh et al., 2004, Journal of Cellular Physiology), in addition to TNFα/RELA regulation of the cytoskeleton (Huber et al., 2004, Journal of Clinical Investigation; Georgouli et al., 2019, Cell). However, there is limited evidence for cytoskeletal regulation of TNF receptors. One example is that the actin-binding protein Filamin interacts with Tumor necrosis factor (TNF) receptor-associated factor 2 (*TRAF2*), which is not itself a TNF receptor but is involved in TNF receptor intracellular signal transduction (Leonardi et al., 2000, Journal of Biological Chemistry). The publishing of these results over 20 years ago emphasises the lack of progression on this topic.

To study cytoskeletal regulation of TNF receptors, we have attempted immunofluorescence visualisation of TNF receptors but found little dynamic range with TNFα treatment. Follow up would require antibody validation or employment of other techniques, which we are interested in but do not believe are necessary for the conclusions of this manuscript.

Reviewer 3 may find of interest that we identified TNFR signalling components as regulated by TNFα in MIA PaCa2 and PANC1 pancreatic cells by RNAseq, which are shown in the revised manuscript (Figure 3C). Notably, we identified that the expression of the gene encoding the signalling adaptor Tumor necrosis factor receptor (TNFR) associated factor 1 *(TRAF1)* is regulated by TNFα in a TNFα dose and duration manner, and *TRAF1* expression is also dependent on RELA nuclear translocation (Figure 3 Cluster 3). We also found that the TNF receptor superfamily members *TNFRSF25* (Figure 3 Cluster 7) and *TNFRSF10B* (Figure 3 Cluster 4) are TNFα-regulated, with *TNFRSF10B* expression dependent on TNFα duration, displaying higher expression at 5 hr TNFα treatments compared to respective 1 hr TNFα treatments. Future experiments could investigate whether these genes are also regulated by modulation of the actin cytoskeleton.

5. The color scheme used in heatmaps (e.g. Figure 2 panels C and I) is problematic because the white is for both the strongest RelA ratio and NAs (missing values). Please use a different color scheme.

Thank you for catching this colour scheme clash. We have omitted heatmaps to display live imaging data from the revised manuscript and checked colour schemes in other figures to avoid confusion over missing values versus quantitative data.

6. Figure 2-supplement 1E seems to show M1-M4 labels mixed up. Please check.

Clustering of single cell RELA-GFP tracks is omitted from the revised manuscript.

7. In line 263 "… correlation between RELA ratio and breast epithelial…", some word seems missing. Does it mean correlation between RELA ratio and "neighbor contact"?

Thank you for identifying this error. We have revised the text:

‘…previously used Bayesian modelling to show that RELA localisation in breast cells is strongly dependent on neighbour contact, cell area, and protrusiveness in the presence and absence of TNFα (Sero et al., 2015).’

8. In line 310, please describe what inhibitor SMIFH2 is at the first mention, for the general readers.

Use of SMIFH2 is omitted from the revised manuscript.

9. In line 313, what does "…for 2 hr then simultaneously with 10 ng/ml TNFalpha for 1 hr." mean? Consecutive treatment or simultaneous co-treament?

Thank you for highlighting the lack of clarity in this treatment schedule. We have rephrased the text accordingly:

‘We generated a single cell dataset for use in Bayesian modelling by treating MIA PaCa2 and PANC1 cells with selected drug doses for 2 hr then simultaneously with TNFα (0, 0.01, 0.1 and 10 ng/ml) for 1 hr…’

10. In line 420, is "NIH-T3" NIH-3T3?

Mention of NIH-3T3 cells is omitted from the revised manuscript.

[Editors’ note: what follows is the authors’ response to the second round of review.]

The manuscript has been improved but there are some remaining issues that need to be addressed, as outlined below:Essential revisions (for the authors):1) Details on the methods used for transcriptional profiling and gene expression analysis should be provided. Reviewers' requests for additional technical clarifications throughout the paper should be carefully considered and addressed where possible.

We have added detailed methods on the sample preparation and RNA-seq processing and bioinformatic analysis:

‘300,000 MIA PaCa2 or PANC1 cells expressing endogenously tagged RELA-GFP and doxycycline-inducible lentiviral IκB-SR (outlined above in ‘Lentiviral cell line generation’) were seeded in t25s to attain 30 % confluence the following day. 24 hr after seeding, cells were treated with 1 µg/ml dox or DMSO. After 48 hr dox, cells were treated with TNFα at a final concentration of 0.01 ng/ml, 0.1 ng/ml or 10 ng/ml, or with a BSA/PBS control for 1 hr or 5 hr. Cells were harvested using the RNeasy Plus mini kit (Qiagen). Two technical replicates were processed per cell line and treatment.

Bulk RNA-seq was carried out using the NovaSeq 6000 (Illumina) with the NovaSeq 6000 Reagent Kit. Samples were processed and reads aligned by the ICR Genomics Facility. The genome was mapped to Hg38 release GRCh38.92. Reads were aligned using Star Aligner version 2.7.6 then processed with HTSeq 0.12.4 to provide counts. Statistical analysis was then-on carried out using the DESeq2 package (Love et al., 2014) in R. Normalised counts from DESeq2 were log_2_ transformed. The log_2_ value for the DMSO control (without dox) for the corresponding cell line (MIA PaCa2 or PANC1) and timepoint (1 hr or 5 hr) was negated from transformed counts. Counts were then z-scored by cell line or across all samples as indicated. Genes with < 10 counts across all samples were removed prior to statistical analysis and genes with 0 counts for any sample were omitted from results.’

2) The method used to quantify "time to first peak" for RELA translocation should be made consistent with similar studies in the field, as noted by Reviewer 2, or should be clarified with explanations of the apparent differences relative to previous literature.

We have re-analysed the ‘first peak’ of nuclear RELA according to previous literature and have updated the figures and text accordingly. Our Methods state the updated method for peak quantification:

‘Nuclear RELA peaks were detected in R by track following Savitzky-Golay filtering using the ‘Signal’ package (filter order p = 3, filter length n = 11) by finding the first timepoint at which the slope of the nuclear RELA curve is ≤ 0.’

3) Inconsistencies between the text and the data shown for expression differences, as noted by Reviewer #2 (point 2) should be rectified or further explained.

We have re-analysed these data and provided a detailed response to Reviewer 2 on the differences noted.

4) Knockdown efficiency for the siRNA experiments should be evaluated with additional data to demonstrate the degree of heterogeneity in knockdown between cells for genes other than RELA. Alternatively, data should be provided to support the assumption that the efficiency of RELA knockdown, at the single-cell level, is representative of the other knocked-down genes.

We appreciate the suggestion to incorporate single-cell analysis as it presents a valuable direction for further study. However, due to constraints on time and resources, we were unable to perform this additional analysis in the current study. Nonetheless, we were able to check induction in 7 out of 8 cases and reduction in all 8 cases by qPCR (Figure 4 – Supplement 1), underscoring the robustness of our experimental approach. The bulk silencing results provide strong validation for on-target depletion, demonstrating that our experiments are technically sound and reliable.

5) Clarification of replicates and statistical significance should be provided for experiments where noted by reviewer 3.

We have updated the figure legends and source data to provide more statistical information in accordance with Reviewer 3’s suggestions. We have also performed a third experimental/biological repeat to provide triplicated data for the live imaging experiment involving siRNA-mediated gene depletions.

Reviewer #1 (Recommendations for the authors):In this revised manuscript, the authors have added substantial new analysis to address the primary critiques of the original manuscript. The authors have made significant changes to their image processing methods and Bayesian network modeling presentation, which in my opinion are adequate to answer the critiques raised on these points. Perhaps the largest change to the manuscript is the inclusion of an RNA-seq experiment and follow-up with knockdowns of genes of interest found in this dataset. The strongest point of this addition is that it is used to investigate the mediators of negative feedback and their differences between the two cell lines analyzed. The RNA-seq data are also used to make a connection between NF-κB activity and actin regulation, which as discussed below, makes less of an impact on the overall conclusions of the paper.Overall, the changes do significantly strengthen the manuscript, but some revisions are still needed to fully integrate the new data.1. By displaying NF-κB localization strictly as nuclear intensity, they avoid the difficulty of using a cytoplasmic ring to calculate nuclear/cytoplasmic intensity, a measurement that can be affected by the cell shape changes induced in their study. However, their description of the method lacks a few details – in particular, is the nuclear intensity calculated as the total RELA signal per nucleus, or the average pixel intensity over the nuclear region (line 567)? Also, the ring method is still used in the fixed cell measurements; it would be helpful to comment in more detail on the previous work that supports the insensitivity of this measurement to cell shape changes (the authors refer to Sero 2015 in their response, but it isn't clear to me where in that manuscript the ring method's response to cell shape is evaluated).

We thank Reviewer 1 for acknowledging the change of our analyses to nuclear RELA intensity from nuc/ring intensity based on the first round of reviewer comments provided. We agree that this is a more straightforward measurement of nuclear RELA signal.

In this manuscript, nuclear RELA intensity is calculated as the mean RELA signal (mean pixel intensity) in the nucleus, i.e. the total RELA signal divided by the area of the nucleus. We have added this definition to the Methods section of the manuscript :

‘Nuclear RELA measurements refer to the mean RELA signal (mean pixel intensity) in the nucleus, i.e. the total RELA signal divided by the area of the nucleus.’

In the Sero et al., (2015) manuscript, interactions between cell shape and the ratio of nuclear:ring region NF-κB intensities are analysed but any alteration to strictly ring region NF-κB intensity with cell shape is not explicitly analysed. Therefore, we have omitted use of the ring region throughout the revised manuscript in accordance with Reviewer 1’s recommendations.

2. The addition of RNA-seq analysis provides a significant amount of new data, and the inclusion of the IkB super repressor is a nice feature of this dataset that helps increase its interpretability. However, overall this addition feels incomplete. There doesn't appear to be any description of the relevant experimental procedures in the methods section. Also, the conclusions from these experiments seem somewhat indistinct. Two actin-regulating genes, NUAK1 and ARHGEF31, are identified as targets of NF-κB, and it is shown that their knockdown modestly changes RELA translocation kinetics. However, this finding doesn't address the question of whether the actin-mediated modulation of NF-κB, which is the focus of the first part of the paper, has a functional role in altering gene expression. Thus, the last section is potentially useful, but a bit confusing in juxtaposition to the first part of the paper.

We have added details of the RNA-seq experiments to the Methods, including the experimental procedure (cell culture and treatment) as well as the sequencing details and analysis.

Our paper concludes that nuclear RELA is linked to heterogeneity in actin features, and that the RELA-regulated genes *NUAK* and *ARHGEF31* play a role in regulating RELA nuclear translocation. We have not assessed downstream transcriptomic changes in the presence of NUAK1 or ARHGEF31 abrogation. However, RELA nuclear translocation has been shown to be strongly associated with target gene expression in multiple studies, including but not limited to:

Ashall L, Horton CA, Nelson DE, Paszek P, Harper CV, Sillitoe K, Ryan S, Spiller DG, Unitt JF, Broomhead DS, Kell DB, Rand DA, Sée V, White MRH. 2009. Pulsatile stimulation determines timing and specificity of NF-κB-dependent transcription. Science 324:242–246. doi:10.1126/science.1164860

Lane K, Valen DV, DeFelice MM, Macklin DN, Kudo T, Jaimovich A, Carr A, Meyer T, Pe’er D, Boutet SC, Covert MW. 2017. Measuring Signaling and RNA-Seq in the Same Cell Links Gene Expression to Dynamic Patterns of NF-κB Activation. Cell Syst 4:458-469.e5. doi:10.1016/j.cels.2017.03.010

Tay S, Hughey JJ, Lee TK, Lipniacki T, Quake SR, Covert MW. 2010. Single-cell NF-κB dynamics reveal digital activation and analog information processing in cells. Nature 466:267–271. doi:10.1038/nature09145

Zambrano S, De Toma I, Piffer A, Bianchi ME, Agresti A. 2016. NF-κB oscillations translate into functionally related patterns of gene expression. eLife 5. doi:10.7554/eLife.09100

We believe that analysing transcriptomic alterations with NUAK1 or ARHGEF31 knockdowns would be a valuable extension to this study but is not necessary for the main conclusion of this paper, which is that actin dynamics and RELA nuclear translocation are interlinked.

Reviewer #2 (Recommendations for the authors):The authors have improved the manuscript extensively both in terms of re-analysis and re-organization of the original data and newly added experimental data and results. I appreciate the effort invested in carrying out such a major revision in response to the original comments from me and other reviewers. Many of my comments have been addressed fully or are no longer relevant in the revised version. The authors performed RNA-seq and GFP-trap experiments partly to address my concern about understanding the core regulatory circuit of NF-kappaB in these PDAC cells. However, some details are missing, and clarifications are needed in several places. While there are numerous points to commend about in this much improved version, in the interest of space, I focus here on remaining issues that need to be addressed before the manuscript can be accepted for publication.New comments:1. The authors note that "the time to peak RELA was highly heterogenous" (line 103). It seems like their definition is different from what other groups have been using. Instead of detecting the *time to the first peak*, it seems like the reported time to peak is catching the *time to maximum value* of nuclear RELA. This is apparent in Figure 1C (easier to see on the right side, e.g. dose response plot of MIA PaCa), where the first peak occurs earlier for the higher TNF dose (which many groups have reported to be valid in multiple cell types. For example, shown by Markus Covert, MH Sung, Michael White, etc). But the panel 1E shows the time to peak is more than 100 minutes (unit is missing in the y axis, by the way), while panel 1C shows a rapid first peak (e.g. < 30 min for 10 ng/ml TNF). I suggest the authors update their definition of time to peak and provide a more accurate quantification of the dose-dependent timing of first peak (well established in the field). With the revised definition, the time to peak may be actually not so heterogenous between individual cells (based on the data spread in panel 1C time course plots). Moreover, the time to peak of 100 minutes is not compatible with the choice of 1 hour TNF treatment in the subsequent immunofluorescence analysis. This needs to be corrected.

We thank reviewer 2 for these comments. The prior submissions of our manuscript defined peak RELA as the maximum nuclear RELA intensity, which is a valid method for quantifying peak RELA, e.g. Sung et al., 2009 (PLOS ONE): ‘The magnitude of first peak response was defined as the maximum nuclear p65 level within 1 hr after TNF-a treatment for each time lapse profile’.

Nonetheless, we have revised our analysis to the first timepoint at which the slope of RELA intensity ≤ 0. We have clarified the method for peak detection in the Methods:

‘Nuclear RELA peaks were detected in R by track following Savitzky-Golay filtering using the ‘Signal’ package (filter order p = 3, filter length n = 11) by finding the first timepoint at which the slope of the nuclear RELA curve is ≤ 0.’

We have updated Figure 1 and Figure 1 Supplement 1 and associated text accordingly:

The change in peak analysis does not change our conclusions about the amplitude of peak RELA at varying TNFα dose but does alter our estimation of the timing of peak nuclear RELA in PDAC cells, which we have revised to a lower timepoint following TNFα addition, in line with Reviewer 2’s suggestion.

We thank the reviewers for pointing out the discrepancy in the initial peak quantification and what is observable from the single cell RELA tracks.

2. There are a few puzzling disconnects between Figure 3 Supplement panels 1A and 1A'-A'. For example, the authors interpret the data in these statements: "NFKB2 is only affected by IκB-SR induction in PANC1 cells (Figure 3 – Supplement 1A).", "NFKB1 is unaffected by RELA-inactivation by IκB-SR in both cell lines", "NFKBIA and NFKBIB were not significantly impacted by RELA inactivation in either cell line, while NFKBIE expression was reduced with IκB-SR in PANC1". But looking at panel 1A, NFKB2 induction is absent in MiaPaCa with IkB-SR; NFKB1 induction is also affected at 5h in both cell lines; NFKBIA induction is off but instead, the constitutive expression is higher (which indicates a high basal transcriptional activity either by NFkB or other TFs like glucocorticoid receptor). Are these simply an outcome of the statistical testing criteria? The discrepancy seems too numerous and pronounced to warrant some double-checking and/or explanation.

Dissimilarities between the reviewer’s observation of the data in Figure 3 Supplement 1A and the statistical results are due to the analysis method used, which relied on collating data from varying doses/timepoints as pseudo-replicates in the absence of biological replicates (detailed in the figure legend), although two technical replicates were processed for RNA-seq per cell line and treatment.

We have reanalysed these data with the Wilcoxon test (previously analysed with a t-test), comparing 5 hr to 1 hr TNFα treatment or the absence and presence of IκB-SR, collating data from 0.01, 0.1 and 10 ng/ml doses in both comparisons. The new statistical significance with the Wilcoxon test and updated text are more consistent with observations by Reviewer 2, including the observation that IκB-SR induction affects *NFKB2* expression:

‘We compared RNA expression in TNFα-treated cells between cells with or without IκB-SR induction (Wilcoxon tests with multiple test correction), which revealed that *RELB*, *REL*, *NFKB2*, and *NFKBIE* have reduced expression with IκB-SR induction in both MIA PaCa2 and PANC1 cells (Figure 3 – Supplement 1A’). These findings indicate that expression of these genes rely on nuclear translocation of RELA, while the other NF-κB and IκB genes, including the canonical RELA binding partner *NFKB1*, do not require RELA activation for expression.

We also considered how expression of NF-κB and IκB genes are affected by the treatment duration of TNFα (Figure 3 – Supplement 1A’’), combining data from 0.01, 0.1 and 10ng/ml TNFα. In both MIA PaCa2 and PANC1 cells, the expression of no NF-κB or IκB genes were statistically different at 1 hr compared to 5 hr TNFα treatment. These results are in line with previous findings that NF-κB and IκB genes are ‘early’ genes upregulated rapidly and consistently following TNFα treatment (Tian et al., 2005; Tay et al., 2010).’

3. How was knockdown efficiency assessed (other than RELA) in the data of Figure 4? Individual cells may show significant heterogeneity in siRNA knockdown (which is different from the RELA siRNA), so I wonder if some of the single cell traces may be from those that didn't have a reduced abundance of the target protein (as well as the mRNA).

We agree that there likely exists heterogeneity in the knockdown of target genes, but we are unable to check this by immunofluorescence due to the quality of available antibodies and discrepancy between RNA and protein abundance. Single cell heterogeneity in siRNA knockdown could certainly be confirmed with single cell RNA-seq, however, this is beyond the financial scope of this project. Nonetheless, we have thoroughly considered Reviewer 2’s feedback and carried out qPCR analysis to confirm that the siRNAs are highly effective in depleting target gene expression with TNFα on the population level:

‘We used non-targeting (NT) siRNA as a negative control for comparison and verified transfection efficacy using RELA siRNA, which abrogated the nuclear RELA signal (Figure 4A), in addition to qRT-PCR (Figure 4 – Supplement 1), which showed reduction of the RNA expression of the targeted gene by each siRNA.’

4. In line 122 "…no differences between cells by cell cycle stage in terms of peak RELA measurements…", it will be good to note that this is in contrast to Michael White's report on cell cycle dependence (Ankers et al. https://elifesciences.org/articles/10473). It is important to note that the crosstalk between cell cycle and NFkB is cell type specific.

It is interesting that our results in PDAC cells differ from those observed in Michael White’s study using HeLa cells. We have updated the Discussion section to reflect these findings:

‘Here, we identified that MIA PaCa2 and PANC1 cells in all cell cycle stages are responsive to 0.1 and 10 ng/ml TNFα, with minimal dissimilarities in RELA translocation responses between cell cycle stages in terms of mean nuclear RELA or the amplitude and timing of peak RELA. In contrast, RELA translocation responses to TNFα were identified in HeLa cells by Ankers et al. (2016) as dependent on the cell cycle phase at the time of TNFα addition in a study using double thymidine block to synchronise cells or Fluorescent Ubiquitination-based Cell Cycle Indicator (FUCCI) labelling to infer cell cycle phase. HeLa cells treated with 10 ng/ml TNFα in late G1 displayed a stronger response than the population average, while S-phase cells showed a suppressed or delayed response. Furthermore, RELA was found to interact with E2F1, a transcription factor regulating the G1 to S transition, in late G1 when E2F1 levels are highest during the cell cycle (Ankers et al., 2016). Conflicting reports of whether TNFα-induced RELA translocation is cell cycle regulated may be due to cancer cell type specific deregulation of cell cycle proteins (Cordon-Cardo, 1995; Otto and Sicinski, 2017) or RELA may not physically interact with E2F1 in PDAC cells in general or specifically in unsynchronised cells.’

5. Regarding the result "Interestingly, the NF-κB protein REL also had reduced interaction with RELA with TNFα.", there is a recent publication (Rahman SMT et al. https://www.cell.com/cell-reports/fulltext/S2211-1247(22)01556-X) reporting that the RelA:c-Rel heterodimer was depleted in the nucleus of TNF-α activated fibroblasts. This reviewer, being the senior author of the study, can't help but find this quite remarkable, and I think that the corroboration might be noteworthy in interpreting the GFP-Trap data.

We are pleased to see that our findings corroborate those published by Rahman et al. and have emphasised these shared results in the Discussion:

‘Co-immunoprecipitation also revealed that the NF-κB protein REL had reduced interaction with RELA with TNFα. This is in line with a recent study (Rahman et al., 2022) that endogenously fluorescently labelled RelA and c-Rel in dual knockin mice and used fluorescence cross-correlation spectroscopy (FFCS) to probe for binding/dimerisation in primary ear fibroblasts to quantify abundance of RelA:c-Rel heterodimers pre- and post-TNFα. Rahman et al., 2022 identified that the relative abundance of the RelA:c-Rel dimer was higher in the nucleus of resting cells then higher in the cytoplasm of TNFα-stimulated cells, despite the individual concentrations of RelA and c-Rel increasing in the nucleus with TNFα. Our combined results suggest that binding of RELA and REL is reduced with TNFα in multiple cell types and in both human and mouse models.’

6. Figure 2: The Bayesian-inferred arrows are sometimes in opposite directions between panel C and panel F (drug treatment data). Any explanations that can help readers understand would be good.

This is a noteworthy aspect of Bayesian networks, which we have elaborated on in the updated Discussion to help interpretation of our networks:

‘Interestingly, the Bayesian-inferred arrows in the RELA network generated from TNFα-treated cells (Figure 2C) sometimes point in opposite directions compared to the network from cells treated with biochemical inhibitors (Figure 2F). This arises from the inherent properties and flexibility of Bayesian networks (Scutari et al., 2018). These networks use a directed acyclic graph (DAG) to represent global probability distributions broken down into smaller local distributions, ensuring no loops or cycles. Despite these constraints, the direction of arrows can change without altering the underlying dependence structure. Multiple valid DAG configurations can exist, so the dependence between two variables A and B can be represented as either A→B or B→A if both are probabilistically equivalent (Heckerman et al., 1995; Scutari, 2010; Scutari et al., 2019). Thus, reversing an arc can leave the overall network structure unchanged, as long as local distributions remain consistent with the data.’

We thank Reviewer 2 for greatly strengthening the manuscript Discussion with their feedback (points 4-6 and previous point 4).

7. Figure 1-Supplement 1 panel A: Please specify that these images are from PCNA.

We have adapted the figure with ‘PCNA-Scarlet’ labels:

8. In Figure 4B legend, something seems to be missing in "(p < value)". In 4C, PCNA Scarlet is supposed to be shown.

Thank you for indicating these errors, we have amended the sentence and updated Figure 4C with PCNA-Scarlet images:

9. Figure 4-Supplement 1: Indicating "siRNA" and "F-actin stain" in this figure would be helpful, even though they are described in the figure legend.

We have removed this figure in response to the feedback by Reviewer 3 (points 10 and 11). Note a new unrelated Figure 4 – Supplement 1 has been added.

10. Figure 2C: In the legend, there seems to be a mix-up of "purple" for "orange" arcs in line 864.

The relevant sentence has been corrected in the Figure 2C legend.

Previous comments that still need attention (in original numbering, followed by additional comments appended after "and"):1. As the authors mentioned in Discussion (lines 370-373 "…, other studies reported more rapid cytoplasmic REL relocalisation…"), the rapid fall of nuclear RelA (around 40 minutes after TNF-α) has widely been reported by several groups (Alexander Hoffmann, Michael White, Markus Covert, Myong-Hee Sung, Savas Tay and their colleagues), but is absent in both PDAC lines. This indicates that the negative feedback genes such as IkappaB α/epsilon or A20 may have very different kinetics if they are induced at all in these cells. It seems important to determine if this is a property of the parental PDAC cells or an acquired feature in the CRISPR knock-in reporter cells during the generation of the two reporter lines. RTqPCR of a few key negative feedback genes in the parental and derived reporter cells, as well as a control cell line (HeLa, MCF10, or THP-1) would be a straightforward way to answer this question. While this is not the focus of the study, I think that understanding which feedback is dysfunctional is probably relevant for therapeutic strategies against PDAC. It would also help the NF-kappaB signaling community understand the common and distinct features of dynamics observed in different cell systems.andIn response to this comment, the authors "carried out RNAseq with MIA PaCa2 and PANC1 cells with endogenously tagged RELA-eGFP". However, no methods are described in the manuscript itself regarding RNA-seq, either in the Methods or figure legends. Therefore, the readers would be left guessing if the samples were from the parental PDAC or the imaged cell lines. This might be an oversight; regardless, authors need to provide the method section on RNA-seq, including sample source, replicates, sequencing platform, data processing and analysis. Also, an accession ID for the RNA-seq data should be provided after depositing the dataset to a public data repository such as GEO, which is a standard requirement, I believe.If the RNA-seq was done on the knockin reporter cells (as stated in the response), then the analysis does not directly address the question I raised in the original comment ("…determine if this is a property of the parental PDAC cells or an acquired feature in the CRISPR knock-in reporter cells during the generation of the two reporter lines"). Barring another round of RNA-seq analysis on the parental cell lines, for a minimum effort, the authors can perform RTqPCR of NFKBIA, NFKBIE, REL, RELB, NFKB2 (those found to be regulated by RelA in the reporter cells) in both the parental and the reporter cells. This would confirm that the observed NF-κB pathway gene expression patterns are indeed a property of PDAC cells.

Thank you for indicating the omission of RNA-seq methodology. We have added detailed information to the updated manuscript on the experimental procedure, sequencing platform and data processing/analysis to the manuscript Methods. We have uploaded the RNA-seq data (FASTQ files and processed counts) to GEO under accession GSE268743. These data will be made immediately available upon publication of this manuscript. To confirm, RNA-seq was carried out using MIA PaCa2 and PANC1 cells with endogenously tagged RELA-GFP.

We acknowledge the reviewer’s insight that the kinetics of NF-κB signalling may be altered in the generation of reporter cell lines. To address the reviewer’s query in differences between RNA expression in parental cells and cells expressing RELA-GFP, we have compared our RNA-seq results in MIA PaCa2 and PANC1 cells expressing RELA-GFP (x-axis) with publicly available RNA abundance data from the Expression Atlas (y-axis) from parental (WT) MIA PaCa2 and PANC1 cells (see Author response image 3). There is a strong correlation (R^2^ = 0.89) between the RNA expression of NF-κB genes, as well as the negative regulators IκB and A20 (*TNFAIP3*), between WT cells and RELA-GFP-expressing cells.

Expression Atlas data obtained from: https://www.ebi.ac.uk/gxa/experiments/E-MTAB-2706We did not have the resources or time to perform new qPCR experiments of these genes directly but believe the increased number of pathway genes compared by RNA seq data (10 genes), and the comparison to parental data from other labs, demonstrate the similarity of both RELA-GFP tagged pancreatic cancer lines to the parental cell lines.

In the literature, it is not generally the case that RNA expression in parental and derived lines are compared. This is in part because it is difficult to separate the effects of fluorescent labels from other confounding factors like genetic drift or epigenetic changes that contribute to clonal heterogeneity in NF-κB signalling, and this level of analysis would require a dedicated study.

Importantly, we validated the function of the NF-κB signalling pathway in our reporter cell lines, responsiveness to TNFα, and used RNA-seq to compare expression of the pathway components to previous studies in the same cell lines. Notably, our RNA-seq analysis was carried out with PDAC cells with endogenously tagged RELA-GFP and we found upregulation of genes encoding IκB proteins with TNFα (Figure 3 – Supplement 1A), which are key negative regulators of RELA.

In addition, we have compared cell proliferation of derived cell lines compared to WT cells, as a major readout of overall function. We identified no difference in proliferation between WT, RELA-GFP and double knock-in (RELA-GFP PCNA-Scarlet) PDAC cells:

**Author response image 3. sa2fig3:** 

We greatly appreciate the feedback and limitations outlined by Reviewer 2 but believe that our study presents the best current understanding of RELA dynamics in PDAC cells.

4. Cytoskeletal structures may influence rates of oligomerization or recycling of the cell surface receptors for TNF-α. Any thoughts on such indirect effects through the upstream signaling events of NF-kappaB activation?andThe author response to this comment contains quite extensive information. The content, or a brief summary, seems to warrant inclusion in the main text of the paper, either in the Results or Discussion. This is an aspect that is not the focus of the study but may be acknowledged as complementary mechanisms to be explored in future studies. Such a bigger picture discussion might encourage readers to explore open topics in their own studies.

We have added an amended version of the previous reviewer response on the cytoskeleton and TNF surface receptors to the manuscript Discussion:

‘One route through which cytoskeletal structures may influence RELA dynamics is by inducing changes in TNF receptor turnover or conformation. In the literature, evidence exists for the modulation of nuclear RELA by actin and tubulin inhibitors (Bourgarel-Rey et al., 2001; Németh et al., 2004), in addition to TNFα/RELA regulation of the cytoskeleton (Georgouli et al., 2019; Huber et al., 2004). However, there is limited evidence for cytoskeletal regulation of TNF receptors. One example is that the actin-binding protein Filamin interacts with Tumor necrosis factor (TNF) receptor-associated factor 2 (*TRAF2*), which is not itself a TNF receptor but is involved in TNF receptor intracellular signal transduction (Leonardi et al., 2000). On a related topic, a recently published study (Alraies et al., 2024) subjected dendritic cells to space confinement and found upregulation of the chemokine receptor Ccr7, in a manner dependent on expression and function of IKKβ and the lipid metabolism enzyme cPLA2. In turn, the prostaglandin E2 receptor (PGE2) receptor, which is in the cPLA2 pathway, induces NF-κB nuclear translocation. This study also identifies a role for ARP2/3 activity in regulating cPLA2 activation via nuclear envelope tensioning. Interestingly, transcriptomics of confined cPLA2 WT and cPLA2 KO DCs revealed correlated expression between Ccr7, the gene encoding IKKβ (Ikbkb), and the major subunits of ARP2/3 (Actr2 and Actr3). Overall, the impact of cytoskeletal

Reviewer #3 (Recommendations for the authors):A revised manuscript presents the study of NF-κB signalling in PDAC tumours. The authors demonstrate that the TNF-induced responses of the canonical p65 signalling are mediated via F- actin dynamics. I believe this represents a novel and important finding.New data in the revised manuscript provide an analysis of TNF-induced gene expression via RNA-seq and identify specific feedback mechanisms, involving IkappaB inhibitors and family members (RelB) as well as actin regulators NUAK2 and ARHGAP31. Furthermore, the authors perform siRNA knockdown experiments to validate specific targets, which provide an excellent contribution to the narrative.The manuscript is presented well and analyses are performed to a high standard. However, the manuscript suffers from some ambiguities regarding sample sizes and statistical analyses. Firstly, most if not all data is not triplicated, instead duplicates while some technical replicates are presented (e.g., Figure 1, Figure 3, Figure 3 S1, Figure 4) -sometimes no information at all is provided (Figure 4C, Figure 4 S1). While part of this info is provided in methods, these should be included in appropriate captions. Secondly, there are limited details in terms of the statistical analyses, typically t-tests are performed (with corrections for multiple testing), often in the case of seemingly small sample sizes. E.g. Figure 3 S1 A' A'- t-test based on 2 samples (as far I understand). T critical he siRNa KO experiment (Figure 4) seems to involve 6 or 8 samples (judging by eye Figure A and B), for which typically a non-parametric test should be used instead. Given that the effect of siRNa-KO (in particular of the actin regulator genes) on p65 dynamics in the live-cell imaging data is subtle (Figure 4A), this poses questions about whether the conclusions are robust.

We are grateful to Reviewer 3 for their positive feedback on our manuscript presentation and analyses, while also bringing to our attention the need for additional statistical information. In response, we have made the following amendments:

Added information on biological and technical (well) replicates to figure legends and source data.Re-analysed Figure 3 Supplement 1 using a Wilcoxon (non-parametric) test and updated the figure and text accordingly.Performed a third biological repeat to provide triplicated data for the live imaging experiment involving siRNAs (Figure 4).Re-analysed Figure 4 using Wilcoxon tests with multiple test correction.Quantified and statistically tested for differences in the area under the curve (AUC) for the live dynamics shown in Figure 4.

Specific comments:1. Line 41: "However, most studies characterising RELA translocation dynamics following stimulation use hyperphysiological TNFα doses (e.g. 10 ng/ml) and exogenous RELA reporters." This sentence is not accurate…both have been studied for more than 10 years, e.g. see [1-3],

We agree with Reviewer 3’s comment and have removed the sentence from the manuscript introduction.

2. Lines 106-112: Authors report correlations between different times, are they suggesting that total p65 is regulated? Please explain or provide the measurement of the total p65 over time.

Figure 1F is not intended to provide insight on the regulation of total RELA over time, but considers how the abundance of nuclear RELA is maintained over time. Our results show that the amount of nuclear RELA, measured by nuclear RELA-GFP intensity, remains persistent following TNFα addition for a long duration (10 hr) for MIA PaCa2 cells, which have a strong correlation (on the single cell level) between nuclear RELA at 300 min versus 600 min. Therefore, MIA PaCa2 cells appear to have particularly long nuclear RELA persistence following TNFα compared to PANC1 cells and quantification in other studies in other cell lines (Ashall et al., 2009; Tay et al., 2010; Zambrano et al., 2016).

3. RELA translocation responses to TNFα are cell cycle independent: Please discuss in the context of the previous work on this subject [4].

We have added to the Discussion comparison of our findings regarding RELA translocation responses to TNFα according to cell cycle stage:

‘Here, we identified that MIA PaCa2 and PANC1 cells in all cell cycle stages are responsive to 0.1 and 10 ng/ml TNFα, with minimal dissimilarities in RELA translocation responses between cell cycle stages in terms of mean nuclear RELA or the amplitude and timing of peak RELA. In contrast, RELA translocation responses to TNFα were identified in HeLa cells by Ankers et al. (2016) as dependent on the cell cycle phase at the time of TNFα addition in a study using double thymidine block to synchronise cells or Fluorescent Ubiquitination-based Cell Cycle Indicator (FUCCI) labelling to infer cell cycle phase. HeLa cells treated with 10 ng/ml TNFα in late G1 displayed a stronger response than the population average, while S-phase cells showed a suppressed or delayed response. Furthermore, RELA was found to interact with E2F1, a transcription factor regulating the G1 to S transition, in late G1 when E2F1 levels are highest during the cell cycle (Ankers et al., 2016). Conflicting reports of whether TNFα-induced RELA translocation is cell cycle regulated may be due to cancer cell type specific deregulation of cell cycle proteins (Cordon-Cardo, 1995; Otto and Sicinski, 2017) or RELA may not physically interact with E2F1 in PDAC cells in general or specifically in unsynchronised cells.’

4. Figure 1 and S1 have no description of statistical tests

We have updated the legends for Figure 1 and Figure 1 Supplement 1 with statistical information – in both cases, t-tests were carried out with Benjamini-Hochberg correction.

5. Figure 2C: numbers characterising the strength of the relationship span over 3 -orders of magnitude. Please comment on their statistical significance…

To generate our Bayesian networks, we used a hybrid algorithm (‘rsmax2’) within the bnlearn R package. This algorithm uses a combination of a score-based method, which is used to evaluate the goodness-of-fit of the entire Bayesian network, and a constraint-based method, which uses statistical tests to assess whether each dependency in the network should be present or absent.

Arc strengths represent the strength of dependencies between variables in the network and a higher arc strength generally indicates:

The parent node has a more substantial impact on the probability distribution of the child node.Higher mutual information between the nodes.The arcs are more likely to be consistently identified across different samples or methods.

Due to the method we used to generate the Bayesian networks in Figure 2C and 2F, all arcs and arc strengths presented are those that are determined as statistically significant, regardless of the actual number. We can therefore use the arc strengths themselves as an indication of predictive power, strength of conditional dependency, degree of mutual information and robustness in structure.

6. 2F. Please provide some validation that chemical perturbation causes measurable changes in any of the cell features…in addition to affecting high-level Bayesian analysis. 2F please provide how many cells analysed

Figure 2 Supplement 3 shows quantification of the effect of chemical perturbation on cell features.

We have added the number of cells for each treatment as additional information in the Figure 2 Source Data provided, in the tab ‘Figure 2 F well and cell frequency’. We have added the following text to the Figure 2 legend:

‘Cells were analysed from three biological repeats, each with 4 wells/technical replicates. Numbers of cells per treatment and cell line are included in the Figure 2 Source Data (range 9800 to 20530 cells per treatment/cell line).’

7. Figure 3 A. How many replicates were assayed, and how the information about different time points is provided on the graph?

We have updated the Figure 3 legend with this information. Two technical replicates per cell line and treatment were processed for RNA sequencing. Statistical analysis of TNFα-regulated genes was carried out by collating RNA abundance at 1 hr and 5 hr treatment:

‘(A) Volcano plot of p-value against mean fold-change (log2) per gene comparing RNA expression with high TNFα (0.1 ng/ml or 10 ng/ml) to control conditions (no TNFα) across MIA PaCa2 and PANC1 cells, using abundance at both 1 hr and 5 hr (handled as pseudo-replicates). Counts were normalised and log2 fold-changes were calculated using DESeq2. Two technical replicates were processed per cell line and treatment.’ (excerpt from Figure 3 Legend)

8. Figure 3 S1 A' A'- Statistical analysis using t-test based on 2 samples (as far I understand). (also line 283 in the text).

We have re-analysed Figure 3 Supplement 1 A’ and A’’ using a Wilcoxon test and have adjusted the figure and text accordingly.

9. Figure 4 "To identify whether siRNAs affected the early/peak RELA and sustained RELA response to TNFα, we calculated the fold change of mean nuclear RELA with each siRNA to NT siRNA at 1 hr or 12 hr TNFα stimulation, and compared fold changes using t-test with multiple comparison correction (Figure 4B and 4C)."The effect of NUAK2 and ARGHAP31 siRNA KO appears to be subtle in live cell imaging from A- is there a statical difference in the AUC or any other characteristics of p65 responses?Analysis in 4B is critical, but not clear to me what is the sample size (how many wells…) and how matching was performed. By eye, it seems that the sample size is between 6-8, but please explicitly provide the number in the legend. T-test with small sample size is not appropriate, a non-parametric test should be used instead.

We thank reviewer 3 for pointing out the incorrect use of the t-test for the analysis in Figure 4B. We have re-analysed all data in Figure 4 and Figure 4 – Supplement 2 using non-parametric tests (Wilcoxon test) with multiple comparison correction and updated the figures and text accordingly. Importantly, we have also carried out a third biological/experimental repeat for the experiment presented in Figure 4 to provide triplicated data.

We have followed the reviewer’s suggestion to calculate the AUC for nuclear RELA signal with the various siRNA-mediated gene knockdowns. We have included this in the main Figure 4 (the new Figure 4B).

‘We considered whether each siRNA affected total nuclear RELA occupancy by quantifying the area under the curve (AUC) of nuclear RELA signal (Figure 4B) for 12 hr control or TNFα treatment. We also quantified the fold-change in RELA AUC (TNFα/control), which informs on how each gene modulates the response of RELA to TNFα. As expected, *RELA* siRNA demonstrated the most significant reduction in the AUC of nuclear RELA signal compared to the non-targeting siRNA in both cell lines in both basal conditions and with TNFα. In both cells lines, depletion of *NFKBIB* also caused significant upregulation RELA responsiveness (fold-change) to TNFα, while PANC1 cells also displayed RELA upregulation with *NFKBIA*, *NFKB2* or *ARHGAP31* depletion. These results suggest that IκBβ plays a significant role in suppressing TNFα induction of nuclear RELA in PDAC cells. Interestingly, nuclear RELA AUC in control conditions was affected by all of the tested NF-κB pathway components in PANC1 cells, indicating that RELA is highly modulated by NF-κB/IκB proteins in the absence of stress or inflammatory cues.’

In addition, we have made the following changes to Figure 4:

Relocated the quantification of nuclear RELA intensity at 1 hr and 12 hr following TNFα from the main figure to Figure 4 – Supplement 2.Updated the Figure 4B and Figure 4 – Supplement 2 legends to clarify the number of experimental and technical replicates presented. To confirm, we performed 3 experimental repeats each with 3 technical (well) replicates.Calculated the fold-change (TNFα/control) for all three nuclear RELA measurements (nuclear intensity at 1 hr post-treatment, nuclear intensity at 12 hr post-treatment, and AUC for nuclear RELA signal) and performed statistical tests.

10. Figure 4C and Supplementary info- no information about data replication -just some images are shown.11. Line 354: MIA PaCa2 and PANC1 cells in the presence of ARHGAP31 siRNA showed flatter morphology and reduction of stress fibre abundance, while NUAK2 siRNA visibly increased actin abundance and the presence of lamellipodia in both cell lines. -> These claims should be statistically tested based on the replicated data.

In response to Reviewer 3’s points 10 and 11, we have removed any claims regarding the effects of siRNA gene knockdown on actin/morphology and have removed the original Figure 4 Supplement 1 (which displayed F-actin for each siRNA) as we are unable to provide quantitative replicated data. We have also updated the Figure 4 legend to include more detail about biological and technical replicates.

12. Line 318: (Figure 4A-D) -there is no D in the figure

Thank you for indicating this error. We have revised the text to state ‘Figure 4A-C’.